# Lineage trajectories and fate determinants of postnatal neural stem cells and ependymal cells in the developing ventricular zone

**Jianqun Zheng**[1⊚]*, **Yawen Chen**[1⊚], **Yukun Hu**[2], **Yujian Zhu**[1], **Jie Lin**[2], **Manlin Xu**[1], **Yunlong Zhang**[2], **Weihong Song**[1]*, **Xi Chen**[2]*

**1** Oujiang Laboratory (Zhejiang Lab for Regenerative Medicine, Vision and Brain Health), Institute of Aging, Key Laboratory of Alzheimer's Disease of Zhejiang Province, The Second Affiliated Hospital, Wenzhou Medical University, Wenzhou, Zhejiang, China, **2** Shenzhen Key Laboratory of Gene Regulation and Systems Biology, Department of Systems Biology, School of Life Sciences, Southern University of Science and Technology, Shenzhen, Guangdong, China

⊚ These authors contributed equally to this work.
* zhengjianqun@ojlab.ac.cn (JZ); weihong@wmu.edu.cn (WS); chenx9@sustech.edu.cn (XC)

## Abstract

The ventricular zone (VZ) harbors the largest neurogenic niche in the adult mammalian brain and is consisted of neural stem cells (NSCs) and multiciliated ependymal cells (EPCs). Previous lineage tracing studies showed that both NSCs and EPCs were derived from radial glial cells (RGCs). However, the transcriptomic dynamics and the molecular mechanisms guiding the cell fate commitment during the differentiation remain poorly understood. In this study, we analyzed the developing VZ of mice at single-cell resolution and identified three distinct cellular states of RGCs: bipotent glial progenitor cells (bGPCs), neonatal NSC-neuroblasts (nNSC-NBs) and neonatal EPCs (nEPCs). The differentiation from bGPCs to nNSC-NBs and nEPCs forms a continuous bifurcating trajectory. Analysis along the NSC branch unveiled a novel intermediate state of cells expressing oligodendrocyte precursor cell (OPC) and neuroblast (NB) marker genes simultaneously. Several transcription factors (TFs) were found to be essential for the EPC-lineage differentiation. Notably, we uncovered that TFEB can tune NSC/EPC bifurcation, independent of its canonical function as a master regulator of the lysosome biogenesis. TFEB activation prevents the overproduction of EPCs by cooperating with LHX2 to balance the expressions of many multicilia-related genes while promotes the differentiation into NSC-NBs. Our results resolve the dynamic repertoire of divergent RGCs during VZ development and offer novel insights into the potential application of TFEB-targeted clinical drugs in VZ-related disorders, such as hydrocephalus and neurodegenerative diseases (NDDs).

**Data availability statement:** Sequencing data are deposited into ArrayExpress and the accession numbers are E-MTAB-13855 (ChIP-seq), E-MTAB-13856 (bulk RNA-seq) and E-MTAB-13858 (scRNA-seq). All other underlying data can be found within the paper and its Supporting Information files. The codes used for data processing, data analysis and graph plotting are available in Zenodo (https://zenodo.org/records/15803123; DOI: https://doi.org/10.5281/zenodo.15803123) and GitHub (https://github.com/sibszheng/VZ_development).

**Funding:** This study was supported by National Key R&D Program of China (https://en.most.gov.cn/, 2021YFF1200900 to X.C), National Natural Science Foundation of China (https://www.nsfc.gov.cn/english/site_1/index.html, 32322019 to X.C), Guangdong Basic and Applied Basic Research Foundation (https://gdstc.gd.gov.cn/, 2023A1515011662 and 2022B1515120077 to X.C), Shenzhen Science and Technology Program (https://stic.sz.gov.cn/, 20220815094330001 to X.C) and the Guangdong Program (https://gdstc.gd.gov.cn/, 2021QN02Y165 to X.C). The funders did not play any role in the study design, data collection and analysis, decision to publish, or preparation of the manuscript.

**Competing interests:** The authors have declared that no competing interests exist.

**Abbreviations:** CSFcerebrospinal fluidEPC, ependymal cells; GOgene ontologyGPC, glial progenitor cells; MWM, Morris water maze; NBneuroblastNDD, neurodegenerative diseases; NSC, neural stem cells; OPC, oligodendrocyte precursor cell; RGC, radial glial cells; RNAiRNA interferencescRNA-seqsingle-cell RNA-sequencingTAP, transit amplifying progenitors; TF, transcription factors; VZ, ventricular zone.

## Introduction

The ventricular system, consisting of inter-connected cavities filled with cerebrospinal fluid (CSF), is a unique characteristic of the brain in vertebrates [1]. The cell layers lining the ventricle are referred to as the ventricular zone (VZ), which is the primary region for neurogenesis and gliogenesis in mammals [2,3]. Mice have a similar ventricular system to humans where four major cavities are observed in both species, making mice an ideal model for the functional and developmental study of the ventricular system.

During the mid-to-late stages of mouse embryonic development, the VZ is predominantly occupied by radial glial cells (RGCs) that are named after their common radial appearance [2]. High heterogeneity was observed during the development of RGCs in the VZ where the majority of RGCs participated in neurogenesis and only one sixth of them contributed to gliogenesis [4]. Around embryonic day 15, the glial-fated RGCs undergo their final mitosis in the embryonic stage, giving rise to glial progenitor cells (GPCs). These GPCs maintain their radial glial morphology until birth [5,6]. In the neonatal stage, some GPCs migrate to the cortex and differentiate into oligodendrocytes and astrocytes, while others remain in the VZ and differentiate into neural stem cells (NSCs) and ependymal cells (EPCs) [7].

NSCs have stem cell characteristics and are capable of self-renewal and differentiation into mature neurons. During the differentiation process, the intermediate transitional states of transit amplifying progenitors (TAPs) and neuroblasts (NBs) can be observed [8–11]. These regenerating features make NSCs with great potential values in treating neurodegenerative diseases (NDDs), brain injuries, and strokes [12]. In the VZ of the adult brain in mammals, NSCs are surrounded by EPCs possessing motile cilia, a microtubule-based organelle protruding from the cell surface [13]. Motile ciliogenesis during EPC differentiation is a multi-step process orchestrated by a tightly regulated transcriptional program. Upstream of the ciliogenesis pathway, GMNC and MCIDAS serve as primary regulators governing centriole amplification [14,15], a process essential for the assembly of multiple cilia. Downstream transcription factors (TFs) FOXJ1 and RFX family members function as direct regulators of core ciliogenic genes [16–19]. MYB and TRP73 have also been identified as critical participants in EPC differentiation process [20,21]. As the apical domain of EPCs is much larger than that of NSCs, the VZ manifests a unique pinwheel organization [22]. The coordinated beating of cilia of EPCs facilitates the circulation of CSF, which is critical for supplying nutrients and removing metabolic waste from brain cells and hence has a great impact on the metabolism, the self-renewal and the differentiation of NSCs [23–26]. Disruption of the circulation of CSF can possibly lead to hydrocephalus development [27].

Although the VZ represents the largest germinal niche in the adult mammalian brain [2,3], the postnatal developmental process of the VZ remains poorly understood. Recent studies with advanced lineage-tracing methods have provided valuable insights on the clonal relationships among cells in the VZ during the development [6,14,28,29]. However, a holistic picture of the transcriptomic dynamics during the

development of the VZ is still lacking, and mechanisms underlying the cell fate commitment during the differentiation remain elusive. Here we performed single-cell RNA-sequencing (scRNA-seq) and TF ChIP-seq to depict the detailed molecule events along the bifurcating differentiation trajectories from GPCs to NSCs and EPCs. Several TFs and pathogenic genes emerged involved in determining the fate of VZ progenitors. Of note, we discovered that TFEB, a master regulator of lysosome biogenesis [30,31], plays pivotal roles in cell fate specification within the VZ, suggesting a novel treatment strategy of targeting TFEB during development for VZ-related disorders such as NDDs.

## Results

### Single-cell transcriptomic data unveil the bifurcating differentiation trajectories of RGCs in the postnatal VZ

To reveal the cellular heterogeneity of RGCs and their differentiation roadmaps to postnatal NSCs and EPCs, scRNA-seq techniques were applied on the developing VZ from mice. The neonatal stage was selected for three practical reasons. First, RGCs stop producing neurons after birth [32], eliminating the influence of neurogenesis. Second, cells responsible for generating oligodendrocytes and astrocytes have migrated to the cortex during this stage [7,28]. Finally, lineage tracing experiments demonstrated that RGCs from the neonatal stage possess the bipotency to differentiate into NSCs and EPCs [14,29].

CD133 has been shown to be specifically expressed on the apical surface of RGCs in the developing VZ [33–35], which serves as an excellent marker for RGC isolation. We set out to examine the CD133 expression pattern and spatial locations of heterogeneous RGCs in VZ at different neonatal stages (postnatal day 0, 5, and 15). Consistent with previous findings, immunohistochemical analysis revealed CD133 enrichment within the 1–2 cell layers adjacent to the lateral ventricle at all examined timepoints (Fig 1A and 1B). Labeling of the ciliary membrane marker (ARL13B) and the ciliary microtubule marker (acetylated-tubulin, ac-TUB) demonstrated dynamic ciliogenesis in CD133-positive RGCs: cells extended singular short primary cilia at postnatal day 0 (P0), formed both primary cilium and multicilia by P5, and manifested dense multicilia by P15 (Fig 1A and 1B). This ciliary transformation pattern suggests a developmental transition from RGCs to mature EPCs. Immunostaining of cell proliferating marker (Ki67) and NSC-lineage marker (ASCL1) indicated that some RGCs had acquired NSC characteristics at P0 (Fig 1A and 1B).

Focusing on early events of EPC-lineage and NSC-lineage differentiation, we collected P0 and P5 samples for scRNA-seq library construction. VZ-containing brain tissues at the two neonatal stages were dissected under a stereoscope and single cell suspension was prepared and stained with an anti-CD133 antibody (see "Methods"). CD133-positive RGCs were isolated using flow cytometry sorting (FACS). Subsequently, scRNA-seq libraries were prepared using the 10× Genomics (10×) platform (Figs 1C and S1A). In addition, small-scale experiments were also performed independently to confirm the findings using a plate-based 3′ scRNA-seq method that was developed previously in our lab [36].

Two biological replicates each of P0 and P5 RGCs using 10× method and two P0 replicates via plate-based method were obtained. A total of 30,445 and 2,594 cells were obtained from 10× and the plate-based methods after quality control (see "Methods"). The data qualities across the six independent experiments were assessed: the Unique Molecular Identifier (UMI) count per cell exceeded 4,500, the average numbers of detected genes ranged from 2,100 to 4,300, and the mitochondrial gene percentage is below 1.5% (S1B Fig). Harmony algorithm [37] was applied to remove batch effect among different replicates (S1C Fig). Unsupervised clustering analysis on the 10× data identified 14 distinct clusters, representing different subtypes of GPCs, NSCs, EPCs and some transitional cell states (Fig 1D). The similar clusters were also observed in the plate-based data (S1D Fig), indicating the robustness of the results. To investigate the relationship among the cells during the development of VZ, a pseudotime analysis was performed using Monocle [38] on the 10× data, and a bifurcating differentiation trajectory was clearly observed (Fig 1E). A very similar pseudotime trajectory with the same bifurcation was also observed (S1F Fig) using the diffusion map algorithm implemented in destiny [39]. In addition, the same results could also be observed in the independently performed plate-based data (S1E and S1F Fig) which

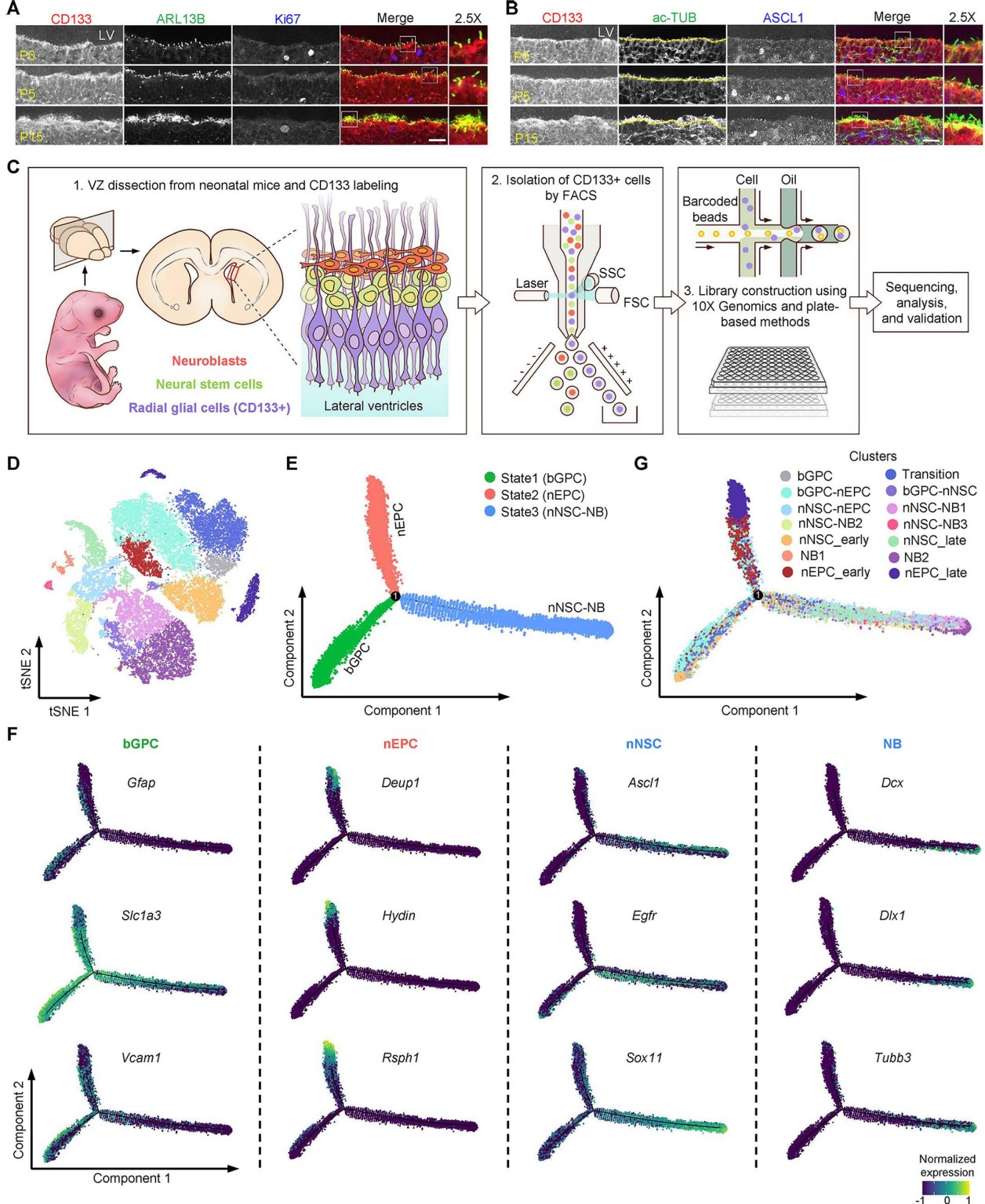

**Fig 1. The developmental trajectory of postnatal ventricular zone by single-cell transcriptomics. (A and B)** CD133 showed specific enrichment on the apical surface of radial glial cells (RGCs) in the developing ventricular zone (VZ) from postnatal day 0 (P0) to P15. Ciliary membrane marker (ARL13B) and the ciliary microtubule marker acetylated-tubulin (ac-TUB) indicated ependymal cell (EPC)-lineage differentiation. Cell proliferating

marker (Ki67) and neural stem cell (NSC)-lineage marker (ASCL1) indicated NSC-lineage transition. The framed region was further magnified to show details. The boundary between lateral ventricle (LV) and VZ was denoted by the yellow dotted line for better distinguishing ciliary ac-TUB and cytoplasmic ac-TUB in **(B)**. The scale bars are 20 μm. **(C)** Schematic diagram of the experimental workflow. RGCs, labeled by CD133, were isolated from the VZ of neonatal mice by fluorescence-activated cell sorting (FACS). Single-cell RNA sequencing (scRNA-seq) libraries were then constructed using droplet-based 10× Genomics (10×) and plate-based modified Smart-seq3 (Plate) method, followed by sequencing, analysis and validation. SSC, side scatter; FSC, forward scatter. **(D)** t-SNE projection of 30,445 cells in the 10× data showing the transcriptional similarities among RGCs at the neonatal stage. Two biological replicates each of P0 and P5 RGCs were performed. **(E)** Pseudotime analysis of all RGCs at the neonatal stage showing a bifurcating differentiation trajectory of the VZ development. Different states are color-coded. The solid black circle indicates the branching point. bGPC, bipotent glial progenitor cell; nEPC, neonatal ependymal cell; nNSC, neonatal neural stem cell; NB, neuroblast. **(F)** Expression profiles of representative markers along the bifurcating trajectory. **(G)** The same bifurcating differentiation trajectory as shown in **(E)**. Cells are colored by the cluster shown in **(D)**.

serves as a validation for the differentiation trajectory. Due to the larger number of cells, we mainly focused on analyzing the 10× data subsequently.

Three main states were observed along the differentiation trajectory (Fig 1E). Many gliogenic marker genes including *Gfap*, *Slc1a3*, *Vcam1*, *Aldh1l1*, *Aqp4*, and *Fabp7* were present in state 1 cells, with the highest expressions observed in cells at the bottom-left corner of the 2D plot (Figs 1E, 1F and S1G). Therefore, it was likely state 1 cells were GPCs and they were the "root" of the differentiation trajectory. High and specific expressions of well-known EPC markers like *Deup1*, *Hydin*, *Rsph1*, *Dynlrb2*, *Ift20*, and *Tekt4* were detected in state 2 cells (Figs 1F and S1G). NSC markers, such as *Ascl1*, *Egfr*, and *Sox11*, showed the highest expression in cells around the middle of the state 3 branch (Fig 1F), and NB marker genes, including *Dcx*, *Dlx1*, and *Tubb3*, were specifically expressed at the end of the state 3 branch (Fig 1F). Previous studies demonstrated that *Gmnc* and *Gmnn* are antagonistic Geminin family members. *Gmnc* expression favored the generation of EPCs while *Gmnn* induced an NSC fate [14,15]. Consistently, *Gmnc* and *Gmnn* showed higher expression in state 2 and state 3 branches, respectively (S1G Fig). Based on the expression patterns of those genes, state 2 and state 3 branches displayed molecular signatures suggestive of the EPC and NSC-NB lineages, respectively.

Next, the 14 cell clusters identified from the clustering analysis were visualized on the pseudotime trajectory (Fig 1G). By looking at the marker genes and the relative positions on the pesudotime, we successfully assigned the identity of each cluster (Fig 1G). The whole pseudotime trajectory described the bifurcating differentiation process from bipotent GPCs (bGPCs) to neonatal EPCs (nEPCs) and neonatal NSCs (nNSCs) and NBs, allowing a detailed analysis of the expression dynamics of various cell fate marker genes along the branches (S1H Fig).

## Molecular cascades underlying bGPC commitment to nNSC-NB

Next, a detailed analysis of the nNSC-NB branch was conducted to identify the genes essential for this lineage. Due to the lack of prior information on the molecular characteristics of nNSCs in the VZ, a robust set of specific markers from adult NSCs was used [8–11]. Adult NSCs include quiescent NSCs (qNSCs) that are in a resting state and activated NSCs (aNSCs) which are in a proliferative state capable of self-renewal and generating neurons under certain conditions [9]. A vast majority of qNSC markers showed higher expression levels in bGPCs and gradually declined during the bGPC to nNSC-NB branch (Fig 2A, top panel). In contrast, aNSC markers showed the opposite trend, where their expressions were low in bGPCs and became progressively higher during the differentiation (Fig 2A, bottom panel). Thus, the conversion from bGPCs into nNSCs was reminiscent of the transition from qNSCs to aNSCs in adulthood. Previous study showed that the VZ growth during juvenile development can be explained by the increasing size of the apical domains of differentiating EPCs, despite a net loss in postnatal NSC number [29]. Based on the available data, it is plausible that bGPCs may evolve into qNSCs in adult stage, while nNSCs likely become adult aNSCs and subsequently migrate away from the VZ (Fig 2B).

To better characterize the transition from bGPC to nNSC-NB branch, differential expression analysis was performed on the four sequential clusters along the pseudotime branch, including nNSC-early, nNSC-late, NB1, and NB2 (Fig 1G). Distinct marker genes were found for each cluster (Fig 2C). Gene ontology (GO) enrichment analysis on the differentially

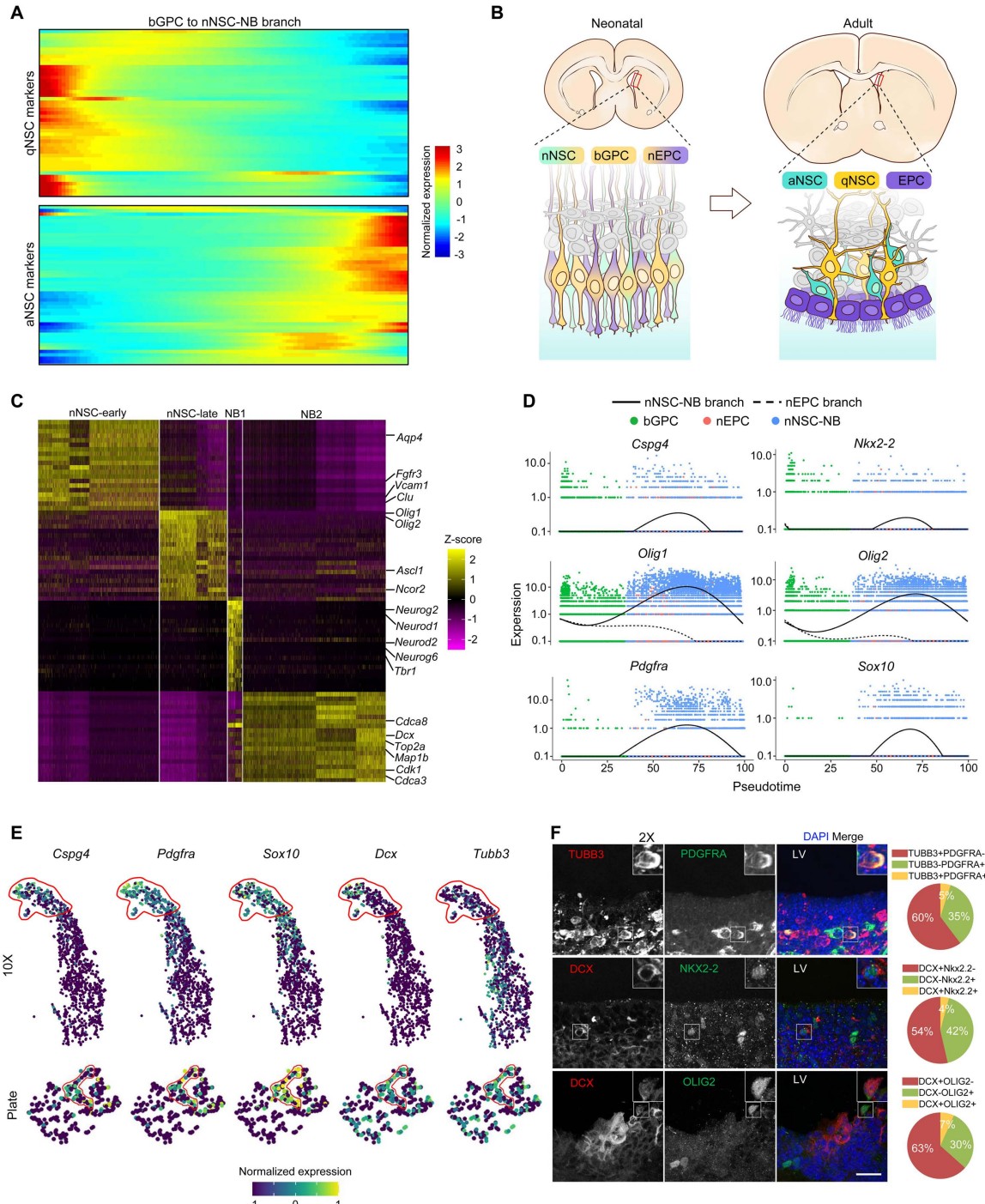

**Fig 2. Characteristics of cells along the nNSC-NB branch. (A)** Heatmaps showing the dynamic expression of marker genes for adult quiescent NSCs (qNSC) and activated NSCs (aNSC) during the differentiation of neonatal bGPCs into nNSCs. **(B)** Schematic diagram illustrating the possible development of the VZ. **(C)** The heatmap of the top 20 signature genes for each of the four clusters along the nNSC-NB branch. Key known marker genes were indicated at the right-hand side. **(D)** Expression dynamics of six well-known marker genes of oligodendrocyte progenitor cells (OPCs) along the pseudotime. **(E)** tSNE visualizations of nNSC_late cells showing co-expressions of OPC markers (*Cspg4*, *Pdgfra*, *Sox10*) and NB markers (*Dcx*, *Tubb3*) from either the 10× or the Plate data. **(F)** OPC-NB bipotent precursors were validated by immunofluorescence of neonatal brain sections. Cells exhibiting positive staining of both the NB marker (TUBB3 or DCX) and the OPC marker (PDGFRA, NKX2-2 or OLIG2) were framed and magnified to show details. Pie charts summarized the percentages of single-positive and double-positive cells. At least 164 cells were quantified in each experiment. LV, lateral ventricle. The scale bar is 20 μm. The data underlying this figure can be found at S1 Data, specifically in the sheet labeled 'Fig 2'.

expressed genes suggests early-stage nNSCs primarily expressed genes related to lipid metabolism and gliogenesis (S2A Fig). By comparison, late-stage nNSCs exhibited high expression of genes associated with ribosome biogenesis and translation (S2A Fig), indicating those cells were preparing for extensive protein synthesis. Interestingly, some late-stage nNSCs expressed markers for oligodendrocyte precursor cells (OPCs) (Figs 2C and S2A), which has not been previously reported during neuronal differentiation [40,41]. This was confirmed by examining the expressions of six OPC marker genes (*Cspg4*, *Nkx2-2*, *Pdgfra*, *Sox10*, *Olig1*, and *Olig2*), all of which had high expression levels within cells around the middle of the nNSC-NB branch (Figs 2D and S2B). Furthermore, some late-stage nNSCs were expressing both OPC markers and NB markers, which was seen not only in the 10× data but also in the plate data (Fig 2E), thereby ruling out the impact of doublets from droplet-based library preparation. Immunofluorescence analysis of brain sections from neonatal mice confirmed the existence of cells co-expressing NB and OPC markers in the VZ, with ratios ranging from 4% to 7% (Figs 2F and S2C). Based on the results, those cells appear to exhibit dual potentials that could give rise to neurons and oligodendrocytes, and hence they were provisionally termed OPC-NB bipotent precursors.

The two NB clusters, NB1 and NB2, were located at the end of the nNSC-NB branch (Fig 1G). TFs controlling neuronal differentiation (such as *Neurod1*, *Neurod2*, *Neurog2*, *Neurog6*) were highly expressed in the NB1 cluster, and NB2 specific genes were mainly involved in cell cycle and cell cytoskeleton (Figs 2C and S2A). By combining those results, the sequential molecular events during the differentiation of the nNSC-NB branch can be described as follows: bGPCs first differentiate into early-stage nNSCs that primarily express gliogenic genes; then cells develop into the late-stage nNSCs when genes related to ribosome production and protein translation are upregulated; Subsequently, cells start expressing TFs facilitating a NB fate and are ready to make morphological changes. Finally, cells enter mitosis, rearrange cytoskeletons and differentiate into NBs. This differentiation process resembles the transition from qNSCs to aNSCs during adulthood [8–11].

## Multiciliogenesis program dominates bGPC transition into nEPC

Next, the nEPC branch from the bifurcating differentiation trajectory was investigated, consisting of early-stage and late-stage EPCs (Fig 1G). Although a set of genes were expressed specifically in the early-stage EPCs, the classic EPC signature genes were only highly expressed in late-stage nEPCs (Fig 3A). GO analysis suggested that genes with higher expression in the early EPCs were mainly associated with gliogenesis and negative regulation of neurogenesis (S3A Fig). Cilia-related genes were low during early EPCs but became highly expressed in late EPCs (S3A Fig).

Defects of motile cilia lead to numerous disorders in humans, including hydrocephalus, situs inversus, infertility and respiratory disorders which are collectively termed motile ciliopathies [42,43]. Currently, 63 genes have been identified to be associated with motile ciliopathies [43]. Among them, 61 were expressed in our dataset and almost all of those genes showed prominent up-regulation along the EPC differentiation branch (Fig 3B).

In comparison to primary cilia, motile cilia possess unique structures composed of the central pair (CP), radial spoke (RS), dynein arm (DA) and nexin link (NL) (Fig 3C) [42,44]. Among the generally recognized 26 CP genes [45], 25 were detected in our data; 15 out of 17 RS genes, 39 out of 40 DA genes and all 11 NL genes [46] were detected in our data (S3B Fig). Almost all of these genes exhibited elevated expression levels during the bGPC-nEPC differentiation (Figs 3D and S3B). The four missing motile cilia structural protein genes were not detected in previously published single-cell transcriptomic datasets from the adult VZ [47] and mass spectrometry data of purified EPC cilia [48] (S3C Fig), confirming their absence in EPC cilia. These results underscore the robustness of our data.

Of note, we observed extensive expression of many structural proteins of motile cilia in bGPCs. As bGPCs differentiated into nNSCs, the expression levels of these proteins gradually declined (Fig 3E). This challenges the notion that structural proteins of motile cilia are exclusively expressed in multiciliated cells. Given EPCs greatly outnumber NSCs in the adult VZ [22], we proposed that some bGPCs maintain ciliary gene expression as a lineage-priming mechanism to bias differentiation towards EPC fates while actively suppressing NSC overproduction.

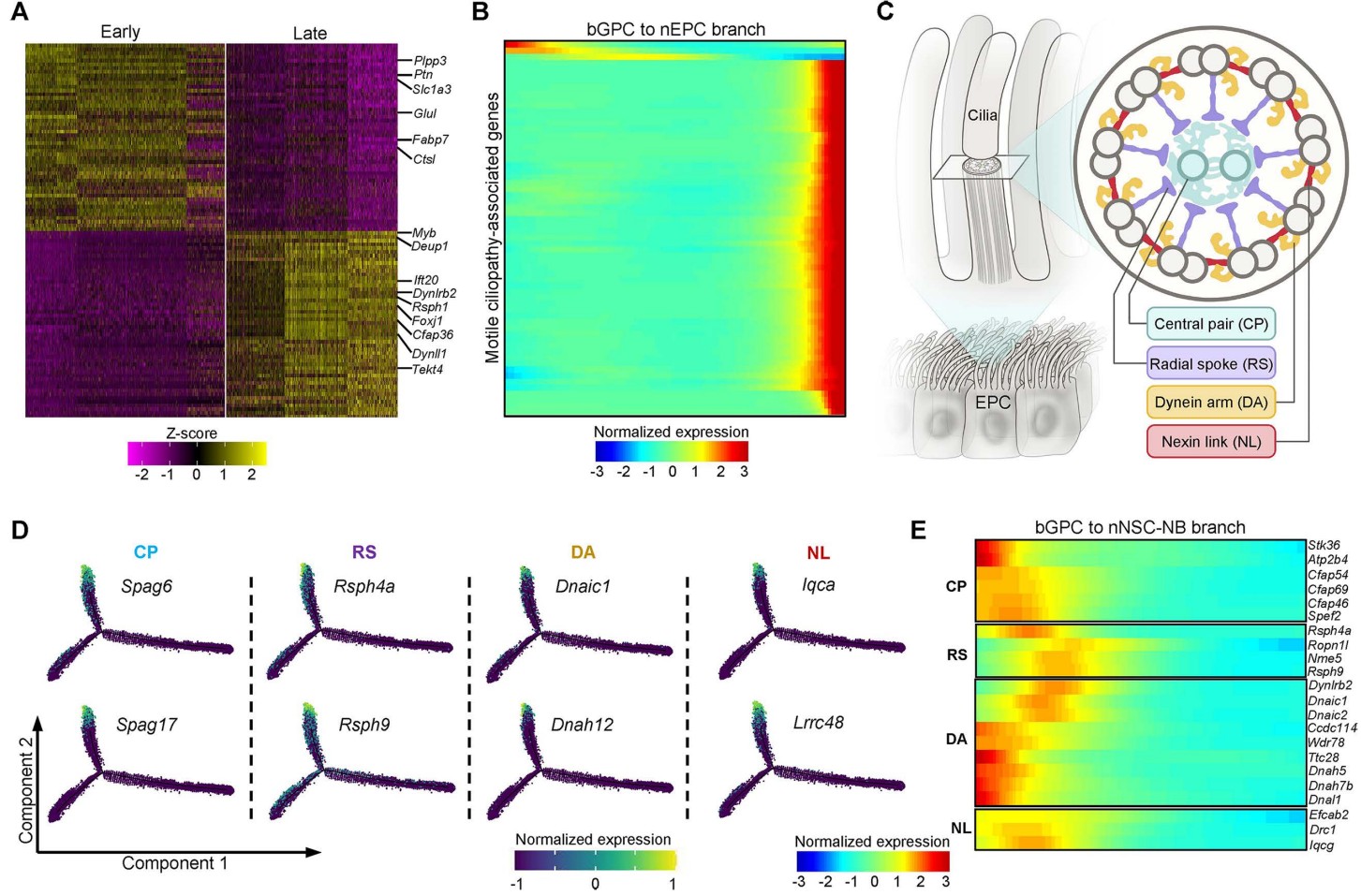

**Fig 3. Features of cells along the nEPC branch. (A)** Heatmap of top 50 marker genes for each of the two EPC clusters. Some key marker genes were indicated at the right-hand side. **(B)** The heatmap showing the dynamic expression of motile ciliopathy-associated genes along the pseudotime trajectory of bGPC to nEPCs. **(C)** Schematic view of the EPC multicilia cross-section. The following key structures are indicated: central pair (CP), radial spoke (RS), dynein arm (DA) and nexin link (NL), all of which are present in multiple motile cilia but not primary cilium. **(D)** The expression profiles of indicated motile-cilia-specific genes along the bifurcating trajectories. **(E)** Heatmap showing the expressions of certain motile-cilia-specific genes are high in bGPCs but become downregulated during the differentiation of bGPCs into nNSC-NBs.

## Differential analysis on the branches unveils hydrocephalus-associated genes impacting EPC differentiation

We further analyzed the differentially expressed genes between the bGPC-nEPC and bGPC-nNSC-NB branches to elucidate how the fate of a bGPC was determined. Among the top 3,000 differentially expressed genes, 1,172 genes were specifically up-regulated in the bGPC-nEPC branch (S4A Fig), and they were termed as nEPC-fate specific genes. GO analysis suggested they were highly enriched in genes related to cilia (S4B Fig), and 303 of them were classified as ciliary genes (S4C Fig). The rest of 869 genes were mainly associated with processes involved in mitochondrial organization or ATP synthesis (S4D Fig), which is likely attributed to the enormous energy demand for cilia motility of EPCs [49,50].

Hydrocephalus is characterized by the enlargement of the brain ventricles caused by obstruction in CSF [27]. To date, more than 400 gene mutations have been identified participating in hydrocephalus development [27,51]. Nevertheless, the mechanisms by which most gene mutations lead to hydrocephalus remain elusive. Due to the fact that hydrocephalus is always accompanied by pathological changes in other brain structures, it is challenging to assess the specific impact of a

single causal factor. The beating of EPC cilia is the main propelling force for CSF circulation in mice, but only mutations in seven genes, including *Foxj1*, *Mcidas*, and *Ccno*, have been identified as causing hydrocephalus by impairing EPC-lineage differentiation [52]. Five of these genes were specifically up-regulated in the nEPC branch (Figs 4A and S4E). Moreover, 86 hydrocephalus-associated genes, previously unreported in the context of ciliogenesis, displayed specific high expression in the nEPC branch (Fig 4B). This implies that hydrocephalus in patients carrying mutations in these genes is attributed to abnormal EPC-branch differentiation in the VZ.

### Transcription factors important for the nEPC fate commitment have distinct expression kinetics in the bifurcating trajectory

Since TF expressions are tightly linked to cell fate determination, we next focused on TFs exhibiting differential expression patterns between the two branches. Among the 278 TFs specifically expressed or up-regulated in nEPCs (Fig 4C), *Foxj1* [16–18,53], *Foxn4* [54], *Rfx2* [18,19], *Rfx3* [19,53], *Myb* [20], *Trp73* [21], *Six3* [55], *Lhx2* [56], and *Id4* [57] have been reported to play roles in EPC production or multiciliogenesis. Here, our data uncovered these known EPC-fate regulators could be classified into two categories based on their expression profiles along the bifurcating trajectory. The first category included *Foxj1*, *Foxn4*, *Myb*, *Rfx2*, and *Trp73*, which showed progressively increasing expression in the nEPC branch with either no expression or decreased expression in the nNSC-NB branch (Figs 4A, 4D, and S4E). The second category included *Id4*, *Lhx2*, *Rfx3*, and *Six3*, which exhibited relatively even or slightly declined expression pattern in the nEPC branch with a more pronounced down-regulation in the nNSC-NB branch (Figs 4E and S4E).

To experimentally verify the expression patterns of the TFs, RGCs were isolated from the neonatal VZ and differentiated into EPCs in vitro under a serum starvation condition (Fig 4F) [48,58]. As progenitor cells give rise to EPCs, the expression levels of the centriole marker γ-tubulin (γ-TUB), the ciliary membrane marker (ARL13B) and the ciliary microtubule marker (ac-TUB) increased, while the expression of the OPC marker OLIG2 declined (Figs 4G and S4F). Thus, the in vitro EPC culture and differentiation system faithfully replicated the in vivo EPC development process. In agreement with our scRNA-seq data, FOXJ1 expression increased as EPC differentiation progressed, while LHX2 showed no evident changes over time (Fig 4G and S4F). These results suggested that EPC-fate regulators fulfil mediating EPC branch output through the potential mechanisms of up-regulating their expression in the EPC branch or down-regulating their expression in the NSC branch.

### *Npas1* and *Foxa2* are indispensable for nEPC-lineage differentiation

To identify novel TFs required for the nEPC-branch differentiation from our scRNA-seq data, candidate TFs were depleted by RNA interference (RNAi) in the in vitro EPC culture and differentiation system (Fig 5A). Multiciliated EPCs were shown to be featured with multiple centrioles or cilia, while other cells contained only two centrioles or one cilium [15,20,22]. Therefore, the status of the EPC differentiation could be assessed by immunofluorescent staining of centriole (γ-TUB) or cilium (ARL13B).

To test whether the in vitro system was applicable for the functional assessment of potential TFs, *Mcidas* and *Lhx2*, two genes with explicit effects on EPCs, were selected as positive controls. Consistent with the previous published works [15,59], depletion of *Mcidas* drastically suppressed multiciliogenesis of EPC precursors (Fig 5B and 5D). In addition, a clear dose-dependent suppression effect of *Mcidas* deficiency was observed. Cells treated with *Mcidas-si2*, which resulted in lower *Mcidas* expression, showed a greater inhibition of multiciliogenesis compared to cells treated with *Mcidas-si1* (Fig 5B and 5D). In contrast to *Mcidas*, the loss of *Lhx2* significantly promoted EPC-lineage specification (Fig 5C and 5D), which agreed with the in vivo results reported earlier [56].

Subsequently, we conducted siRNA transfection to deplete several candidate TFs. We first focused on the *Npas1* gene, also known as neuronal PAS domain protein 1, which was found to suppress the differentiation of cortical neurons [60]. Despite its known function, *Npas1* exhibited specific up-regulation in the nEPC branch within the bifurcating trajectory of

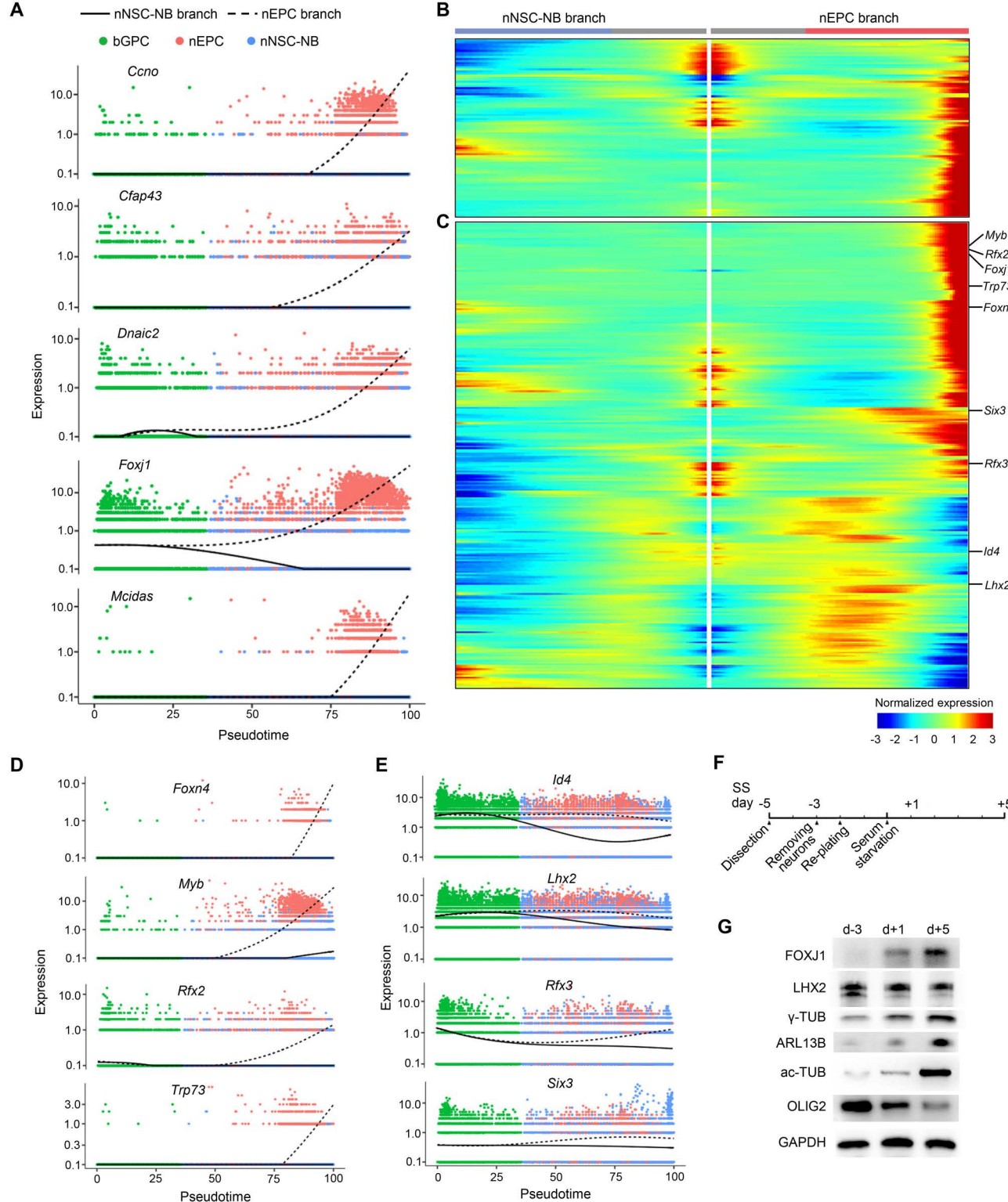

**Fig 4. Expression kinetics of hydrocephalus-related genes and EPC-fate regulators in the bifurcating trajectory. (A)** The five causative hydrocephalus genes were up-regulated specifically in the nEPC branch. **(B)** Heatmap demonstrating 86 genes that have mutations in hydrocephalus were specifically upregulated in the nEPC branch. **(C)** Heatmap showing the dynamic expression of nEPC branch-specific differentially expressed

transcription factors (TFs). The nine well-known TFs that regulate the EPC-lineage differentiation are labelled at the right-hand side. **(D)** The four known EPC-fate regulators showed elevated expressions specifically in the nEPC branch. **(E)** Expressions of the four known EPC-fate regulators exhibited more pronounced downregulation in the nNSC-NB branch than the nEPC branch. **(F)** Experimental scheme for in vitro EPC culture, through serum starvation (SS)-induced differentiation of RGCs from dissected VZ tissues. **(G)** Immunoblotting validation of the expression patterns of FOXJ1, LHX2, and OLIG2 during the in vitro EPC differentiation. Upregulation of the centriole marker γ-tubulin (γ-TUB), the ciliary membrane marker (ARL13B) and the ciliary microtubule marker (ac-TUB) indicate GPC differentiation into EPCs. GAPDH served as a loading control. The quantifications for the immunoblotting were summarized in S4F Fig.

the VZ development (Figs 5E and S5A). Moreover, *Npas1* expression in nEPCs was much higher than in glioblasts, NBs and neurons (S5B Fig), as indicated in the mouse brain development atlas [61], suggesting its involvement in the EPC-lineage differentiation. Deprivation of *Npas1* in the progenitor cells resulted in an evident decrease in the ratio of EPCs from 33.0% to 20.1% (Fig 5F and 5I), suggesting that *Npas1* promotes EPC differentiation.

We next investigated the role of *Foxa2*, a member of the Forkhead-box TF family. Similar to the expression patterns of the other two Forkhead TFs *Foxj1* and *Foxn4* (Fig 4A and 4D), *Foxa2* also exhibited specific up-regulation in the nEPC branch (Figs 5G, S5C and S5D). While *Foxj1* depletion impaired centriole amplification in the early stages of EPC-lineage differentiation (Figs 5J and S5E) and the growth of cilium axoneme in the later stages (Fig 5K), the loss of *Foxn4* loss specifically hampered cilium growth (Fig 5H, 5J, and 5K). These results demonstrated that our data successfully identified genes playing critical roles in bGPC fate specification.

### *Tfeb* restrains the EPC-lineage differentiation downstream of *Gmnc*

Previous studies on the adult VZ have shown that transcription factor EB (TFEB), which modulates lysosome biogenesis, was more abundant in qNSCs compared with aNSCs and facilitated the conversion of qNSCs into aNSCs [62]. In our data, the expression of *Tfeb* in nEPCs was higher than that in bGPCs and nNSCs (Figs 6A, S6A and S6B). Immunoblotting with an anti-TFEB antibody confirmed that the expression of *Tfeb* was progressively elevated as GPCs differentiated towards the EPCs in the in vitro culture system (Fig 6B), indicating potential roles for *Tfeb* in EPCs.

Despite *Tfeb* expression was up-regulated as the nEPC-lineage differentiation progressed, the expression of the lysosomal gene *Lamp1* showed a slight reduction and *Lamp2* stayed relatively unchanged (Fig 6B). Our scRNA-seq data revealed that the expression of 32 out of the 61 lysosomal genes containing the CLEAR motif [30] declined as bGPCs gave rise to nEPCs (S6C Fig). These results imply that lysosomal gene expressions in EPCs, somewhat similar to the situation in embryonic stem cells [63], is independent of *Tfeb*.

To further investigate the role of *Tfeb* during EPC differentiation, two independent siRNAs were designed to deplete *Tfeb* in the in vitro culture system, both of which achieved efficient depletion of *Tfeb* (Fig 6C). Surprisingly, the depletion of *Tfeb* dramatically enhanced EPC differentiation. The examinations on both centriole amplification and cilium formation revealed that the proportion of EPCs increased by nearly two folds (Fig 6D and 6E).

Next we performed bulk RNA-seq analysis on in vitro differentiated EPCs treated with negative control (NC) or *si-Tfeb*. Three independent biological replicates per group were performed (S6D Fig). Overall, there were 884 genes that were significantly up-regulated upon *Tfeb* knockdown and 1,400 genes down-regulated (Fig 6F). Given that the Notch signaling pathway inhibits multiciliogenesis [64], we examined the expression of genes related to this pathway. *Tfeb* loss decreased the expressions of *Notch2*, *Notch3*, and *Hes1* (Fig 6G), all of which are key components of the Notch signalling pathway. In contrast, *Tfeb* deficiency in EPCs did not affect genes related to lysosome formation (Fig 6G). GO analysis of the up-regulated genes revealed that the top five enriched biological processes were all related to cilia (Fig 6H), which agreed with the immunofluorescence results (Fig 6D and 6E). These findings indicate that *Tfeb* deficiency reduces the expression of the Notch signalling pathway and promotes GPC differentiation towards the EPC. Therefore, it is likely that *Tfeb* plays an inhibitory role in EPC production.

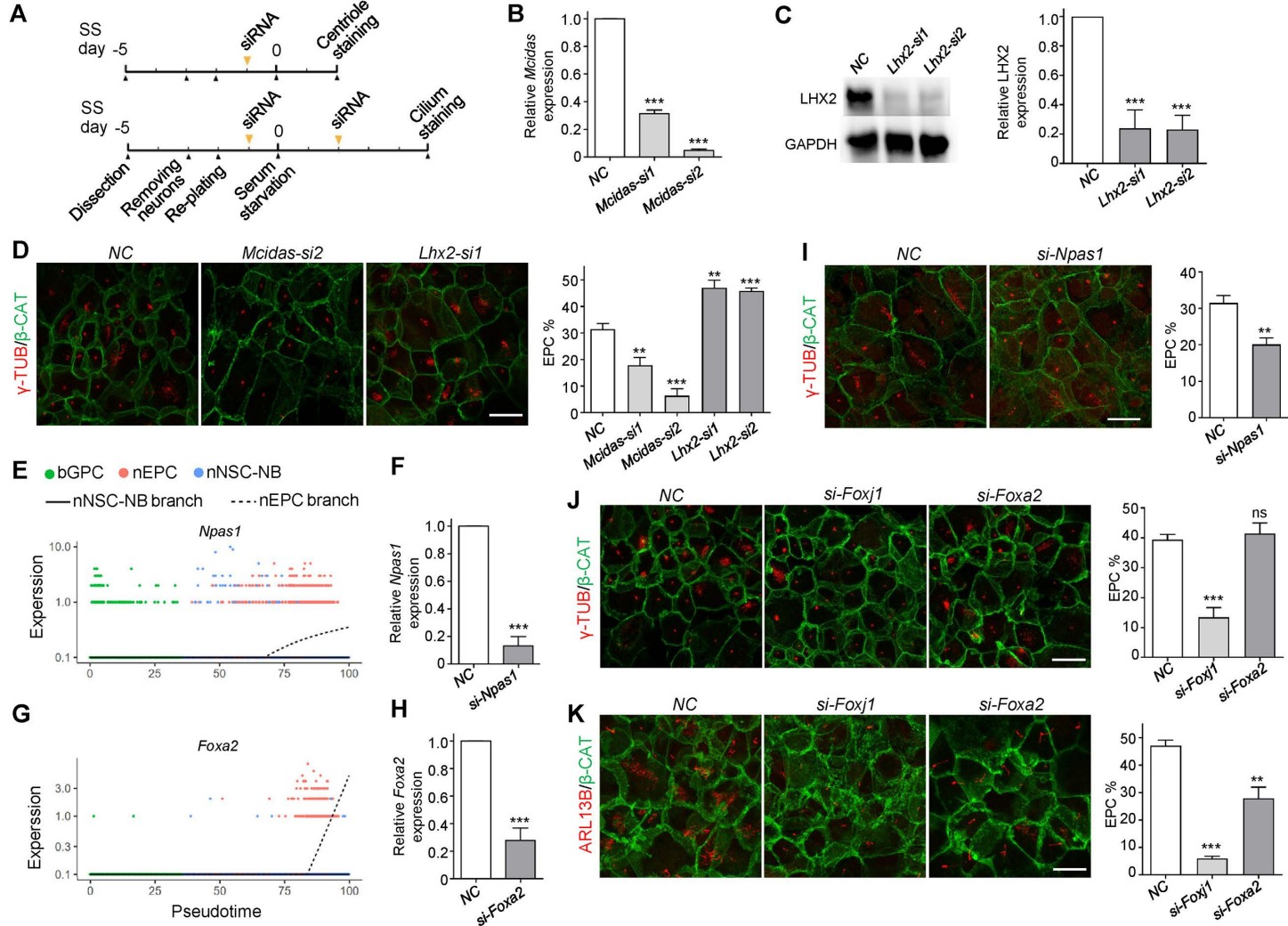

**Fig 5.** *Npas1* and *Foxa2* play roles in the EPC-lineage differentiation. **(A)** Experimental scheme for RNA interference (RNAi) followed by centriole staining or cilium staining. **(B and C)** Quantitative PCR (qPCR) and immunoblotting showed efficient depletion of *Mcidas* and *Lhx2* in EPCs by RNAi. NC, negative control. siRNA transfection was performed at SS d − 1 then cells were harvested for qPCR and immunoblotting at SS d + 2. **(D)** Knockdown of *Mcidas* and *Lhx2* inhibited and promoted the EPC production, respectively. Centrioles were labelled by γ-TUB, and cell borders were labelled by β-catenin (β-CAT). At least 199 cells were quantified in each experiment and condition. **(E and G)** Expressions of *Npas1* and *Foxa2* were specifically upregulated in the nEPC branch. **(F and H)** qPCR showed efficient depletions of *Npas1* and *Foxa2* in EPCs. For *Npas1* knockdown experiment, siRNA transfection was performed at SS d − 1 then cells were harvested at SS d + 2. For *Foxa2* knockdown experiment, siRNA transfection was performed at SS d − 1 and SS d + 2 then cells were harvested at SS d + 5. **(I)** *Npas1* depletion impaired the centriole amplification process during EPC differentiation. At least 257 cells were quantified in each experiment and condition. **(J and K)** Knockdown of *Foxa2* did not affect the centriole amplification process but compromised the cilium formation process during EPC differentiation. The cilium was labelled by ARL13B. FOXJ1, a well-recognized TF essential for cilium formation, served as positive control. At least 172 cells were scored in each experiment and condition. All of the quantification results above were from three independent experiments. Error bars represent the standard deviation (SD). Asterisks indicate *P*-values determined by Student's *t*-tests between NC and siRNA-treated groups, \*\**P* < 0.01; \*\*\**P* < 0.001; ns, not significant. The scale bars are 20 μm. The data underlying this figure can be found at S1 Data, specifically in the sheet labeled 'Fig 5'.

Previous studies have demonstrated that the antagonistic Geminin family members *Gmnn* and *Gmnc* modulate RGC fate decision [14,15]. To investigate the interplay between TFEB and the Geminin members, siRNA-mediated knockdown of *Gmnc* or *Gmnn* was performed in the in vitro culture systems (S7A Fig). In accordance with their known roles [14,15], *Gmnc* deficiency blocked EPC-lineage commitment, whereas *Gmnn* depletion increased the EPC population (S7B Fig).

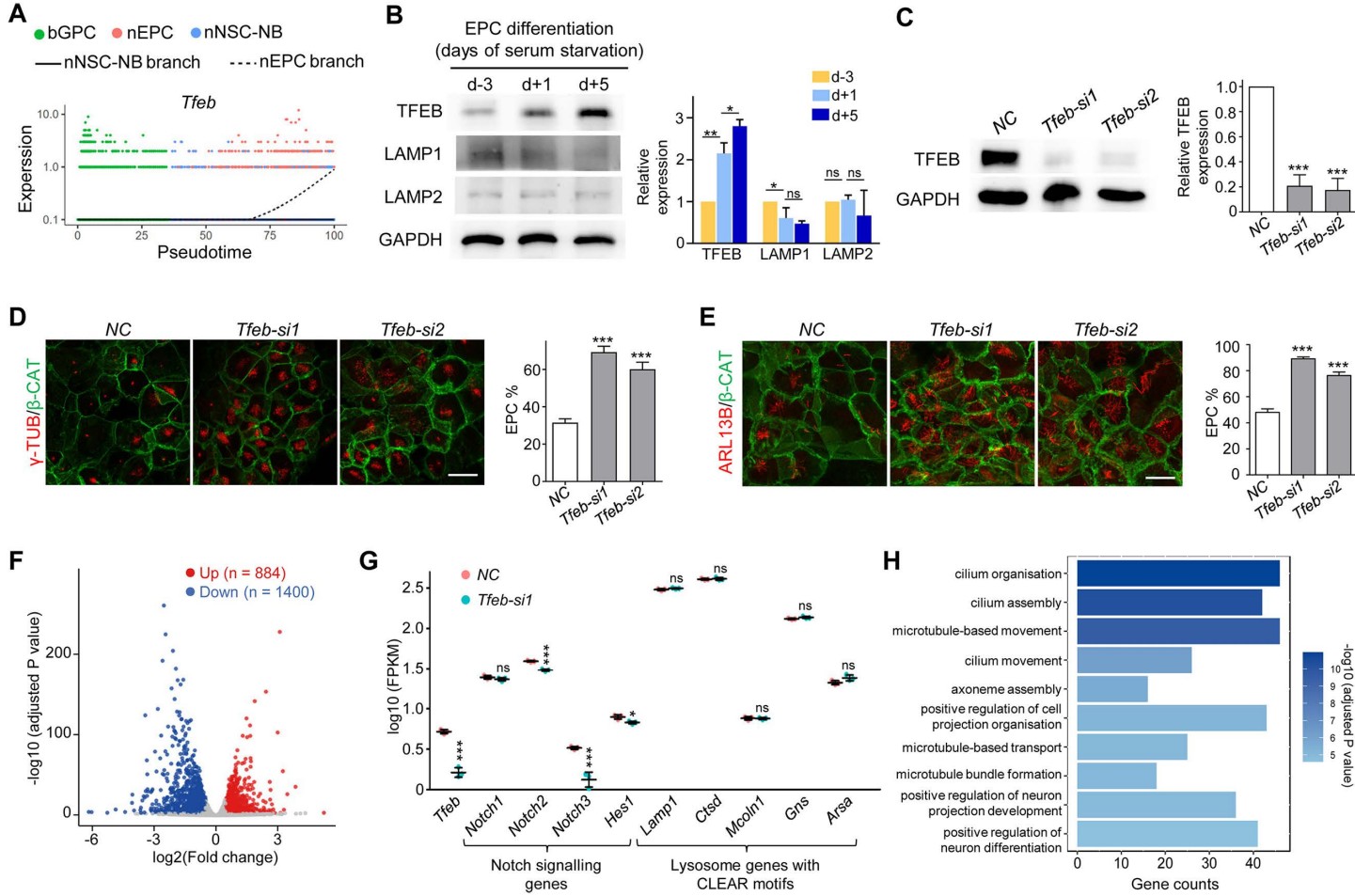

**Fig 6. *Tfeb* restrains the differentiation towards EPCs. (A)** The expression pattern of *Tfeb* along the pseudotime. **(B)** Immunoblotting of TFEB during the in vitro differentiation of GPCs to EPCs. The lysosomal gene LAMP1 and LAMP2 were also shown. GAPDH was used as a loading control. **(C)** Immunoblotting showing efficient depletion of TFEB in EPCs. siRNA transfection was performed at SS d − 1 and SS d + 2 then cells were harvested for immunoblotting at SS d + 5. **(D and E)** Immunofluorescence of γ-TUB, indicating centrioles **(D)**, and ARL13B, indicating cilium **(E)**, upon *Tfeb* depletion showing significantly enhanced EPC-lineage differentiation. β-CAT was used as cell border marker. At least 218 cells were quantified in each experiment and condition. **(F)** Volcano plot of bulk RNA-seq results showing the significant gene expression changes upon *Tfeb* depletion in EPCs. Red and blue dots indicate significantly upregulated and downregulated genes with adjusted *P*-value < 0.001, respectively. siRNA transfection was performed at SS d − 1 and SS d + 2 then cells were harvested at SS d + 5 for RNA-seq library construction. Three biological replicates for *NC* or *Tfeb-si1* treated group were performed. **(G)** Expression levels of indicated Notch signalling genes and lysosome genes with CLEAR motifs from bulk RNA-seq results. **(H)** Top 10 gene ontology (GO) terms of biological processes from upregulated genes upon *Tfeb* depletion. All of the quantification results above were from three independent experiments. Error bars represent SD. Asterisks indicate *P*-values determined by Student's t-tests between NC and siRNA-treated groups, *P < 0.05; **P < 0.01; ***P < 0.001; ns, not significant. The scale bars are 20 μm. The data underlying this figure can be found at S1 Data, specifically in the sheet labeled 'Fig 6'.

QPCR analysis showed that *Gmnc* ablation caused a 73.5% reduction in *Tfeb* expression, while the loss of *Gmnn* exerted no significant effect (S7C Fig). This finding is consistent with a recent published dataset displaying that GMNC overexpression induced upregulated expression of *Tfeb* [65]. On the other hand, the loss of TFEB did not impact *Gmnc* expression and slightly reduced *Gmnn* expression by 17.8% (S7D Fig). Moreover, co-immunoprecipitation assays revealed the absence of direct physical interactions between TFEB and either Geminin protein (S7E Fig). These results suggest that TFEB undergoes indirect modulation by GMNC during multiciliogenesis.

## TFEB activation blocks GPC specification into EPCs through suppressing multicilia-related genes

We next utilized other independent approaches to investigate how *Tfeb* functions in the process of EPC-linage specification. Previous studies have demonstrated that high concentrations of trehalose or sucrose induce lysosome stress, leading to the nuclear translocation and hence the activation of TFEB [30,66]. Corroborative with the results from the siRNA experiments, 100 mM trehalose or sucrose induced nuclear localization of TFEB in GPCs (Figs 7A and S8A), almost abolishing their differentiation into EPCs (Figs 7B and S8B). When GPCs were treated with varying concentrations of trehalose, we observed a strong negative relation between TFEB nuclear translocation and the differentiation efficiency of GPCs into EPCs (Fig 7A and 7B). At a concentration of 10 mM trehalose, the nuclear translocation of TFEB could already be observed in many cells (Fig 7A). Interestingly, the nuclear localization of TFEB and the expression of FOXJ1 were mutually exclusive, where TFEB-positive nuclei exhibited no FOXJ1 protein (67%) and TFEB-negative nucleus showed positive FOXJ1 staining (30%) (Fig 7C). The results confirmed the inhibitory effect of TFEB activation on EPC-lineage differentiation. In addition, we found that amino acid starvation also impeded EPC-branch differentiation by inducing TFEB nuclear localization (S8C and S8D Fig).

Prior studies have identified that mTOR phosphorylates TFEB to prevent its activation [67,68], while calcineurin dephosphorylates TFEB to facilitate its activation [69]. Treatment of GPCs with Torin1, an mTOR inhibitor [67,68], enhanced TFEB nuclear translocation and hampered EPC differentiation (Fig 7D and 7E). On the other hand, treatment with cyclosporin A (CsA), a calcineurin inhibitor [69], slightly increased the efficiency of EPC differentiation (Fig 7E), though CsA was unable to induce the nuclear translocation of TFEB (Fig 7D). These results implicate that the inhibitory effect of TFEB during EPC-lineage differentiation is closely associated with its phosphorylation.

To further elucidate the molecular mechanisms underlying the role of TFEB in EPC-lineage specification, chromatin immunoprecipitation followed by sequencing (ChIP-seq) was conducted in in vitro differentiated EPCs. We identified a total of 10,646 TFEB binding sites in the genome (S8E Fig). De novo Motif analysis returned the top motif that resembled the TFE family motif (Fig 7F), confirming the high quality of the data. Interestingly, the RFX family motif and the LIM-HD family motif were also found enriched within the TFEB binding sites (Fig 7F). The RFX transcription factors were previously shown to be essential for ciliogenesis [18,19,53], and LHX2, a member in the LIM-HD family, was implicated in multiciliogenesis suppression (Fig 5D) [56]. It is likely that TFEB cooperates with LHX2 to block the expression of multicilia-related genes. To test this hypothesis, immunoprecipitation showed that both endogenous TFEB in EPCs and exogenous TFEB overexpressed in HEK293T cells interacted with LHX2 (Fig 7G). Moreover, TFEB bound to the gene body of *Foxj1* according to our ChIP-seq data (Fig 7H), and knockdown of *Tfeb* resulted in elevated expression of *Foxj1* (Fig 7I). These results indicated that the binding of TFEB on *Foxj1* might have an inhibitory effect, which explains the immunofluorescence result showing that nuclear translocation of TFEB and FOXJ1 expression were mutually exclusive (Fig 7C). Furthermore, TFEB was found to bind to the gene bodies of other cilia-related genes, including *Dnaic1*, *Anxa5*, and *Wdr63*, and the binding also seemed to have suppressive effects on their expressions (Fig 7H and 7I).

## Overexpression of *Tfeb* induces NSC-lineage differentiation

To exclude the off-target effects of RNAi, an siRNA refractory version of *GFP-Tfeb* was introduced to *si-Tfeb* treated EPCs. The expression of GFP-TFEB significantly mitigated the EPC overproduction caused by the loss of TFEB, as compared to cells expressing GFP only (Fig 8A and 8A'). Of note, some GFP-TFEB-positive cells manifested neuron-like morphologies (Fig 8A and 8A'). When GFP-TFEB was overexpressed in wild-type GPCs, 35.4% of them displayed neuron-like morphologies, and only 19.2% gave rise to EPCs (Fig 8B and 8B'). In comparison, nearly half of the cells transfected with only GFP differentiated into EPCs, and neuron-like morphologies were rarely observed (Fig 8B and 8B'). These findings suggest that an excess of TFEB promotes the differentiation of bGPCs towards the nNSC-NB fate.

To probe whether the effects of TFEB overexpression are associated with its transcriptional activity, several constitutively active mutants of TFEB [70] were introduced to GPCs. The mutant GFP-TFEB (S210A) showed the most

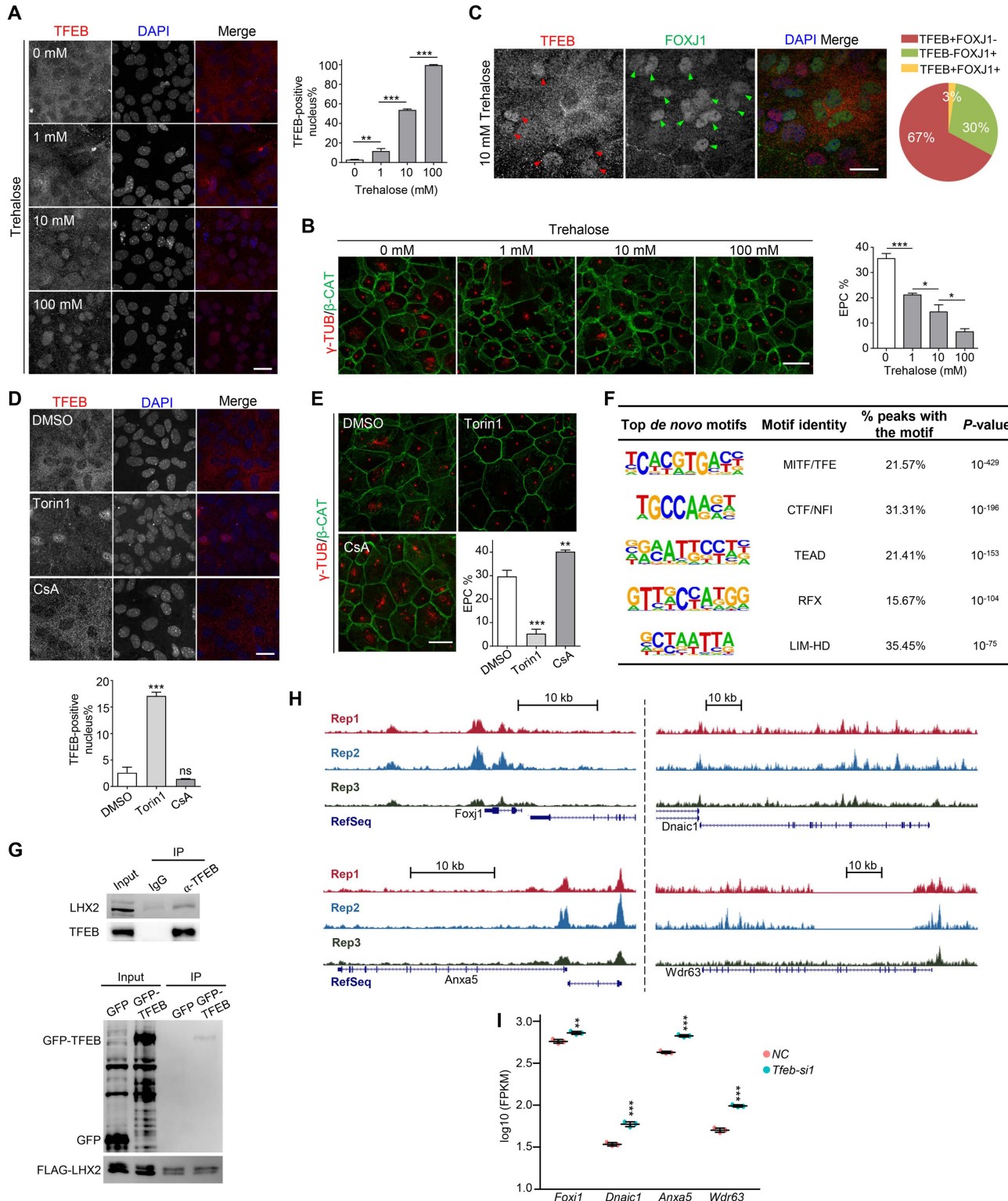

**Fig 7. Activated TFEB cooperates with LHX2 to inhibit multiciliogenesis. (A)** Nuclear localization of TFEB by immunofluorescence in the EPC culture system showed a dosage-dependence on trehalose treatment. At least 151 cells were quantified in each experiment and condition. **(B)** Immuno-fluorescence of γ-TUB showing the EPC percentage was negatively correlated with the concentrations of trehalose in the culture media. β-CAT was used

as cell border marker. At least 233 cells were quantified in each experiment and condition. **(C)** Upon 10 mM trehalose treatment, immunofluorescence analyses showing TFEB and FOXJ1 displayed mutually exclusive nuclear localization in glial cells. Red and green arrowheads denote nuclear localizations of TFEB and FOXJ1, respectively. The pie chart summarized the percentages of single-positive and double-positive cells. A total of 329 cells were quantified. **(D)** Immunofluorescence analyses showing the translocation of TFEB from the cytosol to the nucleus upon 250 nM Torin1 (mTOR inhibitor) treatment. TFEB remained in the cytoplasm upon 4 μM CsA (Calcineurin inhibitor) treatment. CsA, cyclosporin A. At least 288 cells were quantified in each experiment and condition. **(E)** Immunofluorescence of γ-TUB showing treatment with Torin1 and CsA impeded and facilitated EPC-lineage differentiation respectively. β-CAT was used as cell border marker. At least 223 cells were quantified in each experiment and condition. **(F)** Top 5 enriched motifs within TFEB-bound regions identified by ChIP-seq. Cells were harvested at SS d+5 for ChIP-seq library construction. Three biological replicates were performed. **(G)** Co-immunoprecipitation (IP) revealed the interaction between TFEB and LHX2 in EPCs (top panel) and HEK293T cells (bottom panel). IgG and GFP were used as negative controls. SS d+5 EPCs were lysed and subjected to endogenous TFEB IP. FLAG-LHX2 combined with GFP or GFP-TFEB were expressed in HEK293T cells for 36 h and then harvested for FLAG IP. Input samples, 10% and IP samples, 50% were subjected for immunoblotting analysis. **(H)** UCSC genome browser tracks showing the TFEB bing peaks around four representative ciliary gene loci *Foxj1*, *Dnaic1*, *Anxa5*, *and Wdr63*. **(I)** Expression levels of four indicated ciliary genes from bulk RNA-seq results. siRNA transfection was performed at SS d − 1 and SS d+2 then cells were harvested at SS d+5 for RNA-seq library construction. Error bars represent SD. Student's *t*-tests were performed between DMSO and drug-treated groups in **(D, E)**. Asterisks indicate *P*-values, *$P < 0.05$; **$P < 0.01$; ***$P < 0.001$; ns, not significant. The scale bars are 20 μm. The data underlying this figure can be found at S1 Data, specifically in the sheet labeled 'Fig 7'.

pronounced nuclear localization (S8F Fig). The Ser210 in mouse TFEB was shown to be a conserved residue previously identified as an mTORC1 target [67,68]. Cells expressing GFP-TFEB (S210A) exhibited a remarkably higher proportion (67.2%) of neuron-like morphologies, and only 3.1% gave rise to EPCs (Fig 8B and 8B'). In contrast, cells expressing GFP-TFEB-ΔNLS, a mutant of TFEB in which the nuclear localization signal (NLS) (amino acids 244–247) was mutated to alanines [67], showed a much lower proportion (5.9%) of neuron-like morphologies and a higher differentiation efficiency (35.3%) into EPCs (Fig 8B and 8B').

To verify that *Tfeb* overexpression indeed promotes the differentiation towards NSCs, RGCs were isolated and cultured in vitro in the presence of serum and immunostained with a TUBB3 antibody. A 39.4% of GFP-TFEB-positive cells and 30.9% of GFP-TFEB (S210A)-positive cells were differentiated into neurons (Fig 8C and 8C'). In contrast, the ratio fell to only 6.4% and 11.9% in GFP-positive and GFP-TFEB-ΔNLS-positive cells, respectively (Fig 8C and 8C'). In addition, 33.7% of GFP-TFEB-positive cells and 48.8% of GFP-TFEB (S210A)-positive cells displayed neuron-like morphologies without expressing TUBB3 (referred to as neuron-like cells), compared to only 11.3% in GFP-positive and 10.3% in GFP-TFEB-ΔNLS-positive cells (Fig 8C and 8C'). Hence the TFEB activation in neonatal VZ facilitates bGPC specification towards the nNSC-NB lineage.

Collectively, our findings indicate that TFEB controls neonatal EPC/NSC bifurcation. Activated TFEB, on one hand, prevents the overproduction of EPCs by interacting with LHX2 and jointly binding to the regulatory regions of ciliary genes such as *Foxj1* to suppress their expressions. On the other hand, activated TFEB promotes the differentiation of bGPCs into nNSCs, a process that can be modulated via the phosphorylation on the S210 residue by mTORC1 (Fig 8D).

## Administration of TFEB-targeted clinical drug during development mitigates memory defects of a NDD mouse model

NSC number and activity decrease during pathogenesis of NDDs [71,72] and numerous preclinical studies have shown therapeutic benefits of NSC transplantation in NDD models [12,73]. Based on our finding that TFEB can tune NSC/EPC bifurcation in the developing VZ, we investigated whether targeting TFEB at developmental stage could mitigate NDD symptoms. Consistent with previous finding that the clinical drug Rapamycin activates TFEB [74,75], Rapamycin treatment induced TFEB nuclear translocation in GPCs (S9A Fig). We subsequently administered Rapamycin to 5 × FAD mice, a validated familial Alzheimer's disease model [76], from birth onward (see "Methods"). At 3 months of age, spatial learning and memory were assessed using the Morris water maze (MWM). Visible platform testing revealed comparable motor function and visual acuity among 5 × FAD, Rapamycin-treated 5 × FAD, and wildtype groups, eliminating confounding sensory-motor factors in memory assessment (S9B and S9C Fig). During training, no intergroup differences emerged in

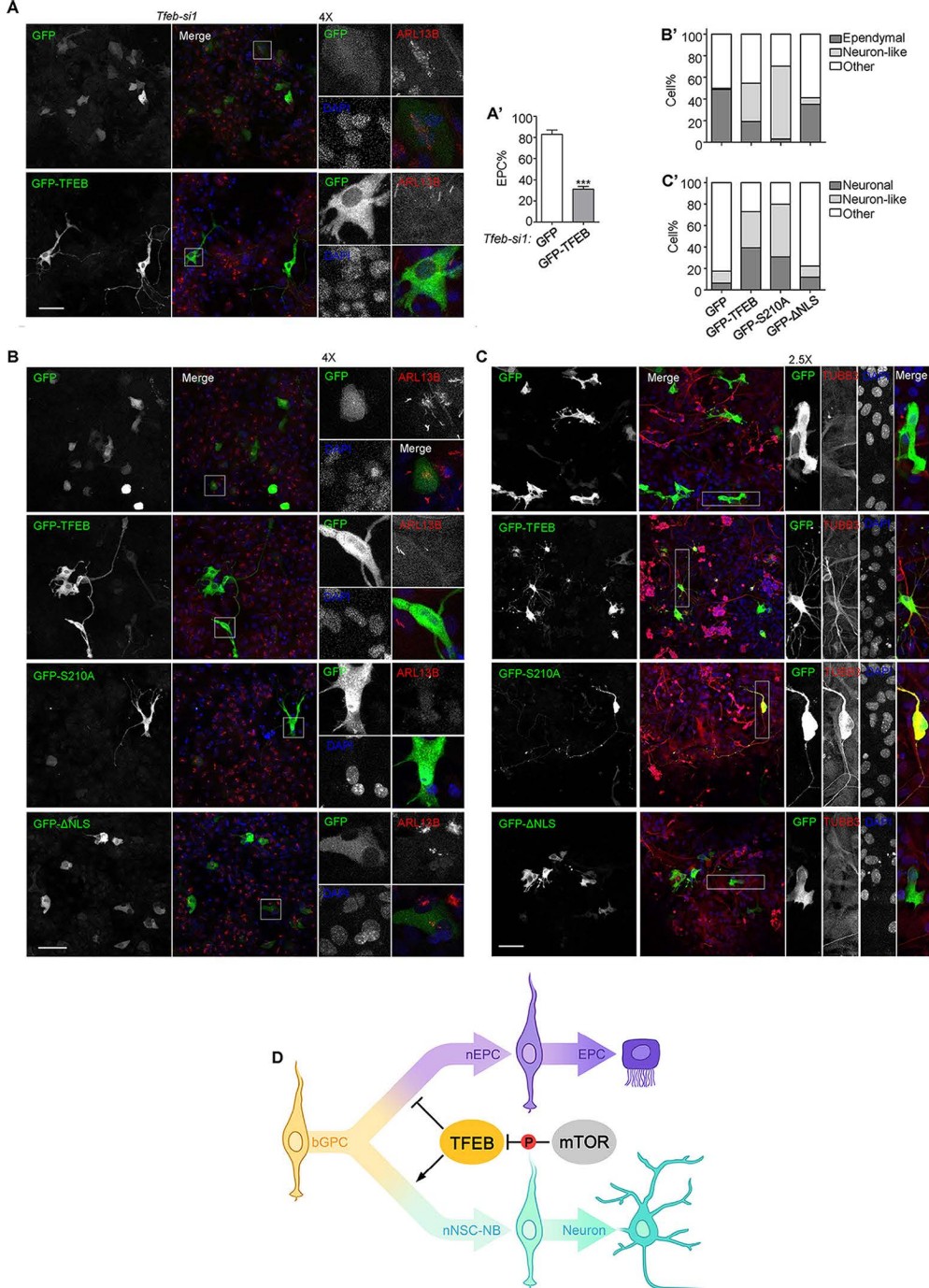

**Fig 8. Overexpression of TFEB induces a NSC fate. (A** and **A')** Immunofluorescence of ARL13B showing the overexpression of an RNAi-resistant GFP-TFEB in *Tfeb-si1*-treated GPCs rescued overproduction of EPCs and induced the generation of cells exhibiting neuron-like morphology. Cells were transfected at SS d − 1 to express GFP-TFEB or GFP on top of the transfection of *Tfeb-si1* and harvested for immunofluorescence at SS d + 5. GFP was used as a negative control. The framed region was further magnified to show details. The quantification results were from three independent experiments. At least 70 cells were scored in each experiment and condition. Error bars represent SD. Asterisks indicate *P*-values determined by Student's *t*-tests between the two groups, ***P* < 0.001. **(B** and **B')** Immunofluorescence of ARL13B showing the overexpression of GFP-TFEB and TFEB-S210A, but not TFEB-ΔNLS induced production of cells exhibiting neuron-like morphology and blocked EPC-lineage differentiation. Plasmid transfection was performed at SS d − 1 and then cells were harvested for immunofluorescence at SS d + 5. At least 64 cells were scored in each condition. **(C** and **C')** Expression of GFP-TFEB and

TFEB-S210A, but not TFEB-ΔNLS induced the conversion of GPCs into neurons. Cells were cultured in the presence of serum. Neurons were labelled by TUBB3 staining. At least 104 cells were scored in each condition. The scale bars are 50 μm. **(D)** A model illustrating how TFEB functions in the process of bGPC specification during VZ development. The data underlying this figure can be found at S1 Data, specifically in the sheet labeled 'Fig 8'.

latency to locate the hidden platform (S9D Fig). Probe trial analysis revealed memory deficits in 5 × FAD mice, evidenced by reduced platform crossover number and the diminished platform tendency in movement heatmap (S9E and S9F Fig). Notably, Rapamycin treatment significantly rescued the performance of memory impairment (S9E and S9F Fig). These findings suggest that developmental-stage administration of Rapamycin ameliorates spatial memory deficits in NDD model, showing the potential application of our study for VZ-related disorders.

## Discussion

### Unveiling the co-differentiation lineage roadmaps of postnatal EPCs and NSCs

Previous transcriptomics studies on postnatal NSCs have predominantly focused on the transition of qNSCs to aNSCs in the adulthood [8–11], without defining the developmental origins of these two NSC subtypes and their primary niche mature EPCs. Through single-cell transcriptomic profiling of the VZ in neonatal mouse brains, we identified distinct progenitor states and mapped their differentiation trajectories. Our analysis revealed a developmental bifurcation in which bGPCs diverge into nNSC and nEPC lineages. By integrating our data with the transcriptomic profiles of NSCs at the adult stage, we extrapolated that neonatal bGPCs become qNSCs in adulthood, while nNSCs evolve into aNSCs (Fig 2B). These findings establish a developmental continuum between neonatal and adult NSCs and highlight the shared ontogeny of EPCs and NSCs.

Our data demonstrated that the developmental processes of postnatal EPCs and NSCs are connected. For NSC-lineage specification, the expression of ciliary structural proteins was down-regulated, whereas EPC-lineage differentiation was dominated by transcriptional programs of these structural proteins. Notably, key TFs for multiciliogenesis showed divergent expression patterns between lineages. These findings unveil the diverse gene regulatory mechanisms for the cell fate decision exploited by bipotent progenitors during their differentiation processes.

### Identification of OPC-NB bipotent precursors

It has been previously assumed that OPCs and NBs differentiate along separate lineages [40]. Our work identified a transitional progenitor population co-expressing markers of both lineages. This observation suggests the existence of a bipotent progenitor cell state in the VZ capable of participating in both oligodendrocyte and neuronal differentiation. Future studies should focus on verifying the differentiation capacity of these OPC-NB bipotent precursors. Given that oligodendrocytes are responsible for ensheathing axons to ensure the smooth conduction of action potentials [77], our findings point to the possibility of new avenues for regenerative approaches to restore myelin integrity and promote functional recovery in cases of neural injuries. By manipulating the fate determination of these bipotent precursors, it may be possible to enhance the production of both oligodendrocytes and neurons, leading to improved repair outcomes.

### A resource for systematic identification of fate determinants for bGPC differentiation

Through differential analysis of gene expressions in the two branches from the bifurcating trajectory, we identified more than 200 TFs that have not been characterized in the context of neonatal gliogeneis may play roles in fate determination towards EPCs (Fig 4C). Experimental validations revealed that several TFs play roles in EPC differentiation by modulating ciliogenesis. Interestingly, *Tfeb* emerged as a key player in both EPC-lineage and NSC-lineage specifications, governing the fate of bGPCs (Fig 8D). These results demonstrate the power of our scRNA-seq data and the potential of this resource for understanding the molecular mechanisms underlying bGPC fate determination. Further experimental

PLOS Biology

validation of other potential TFs may help understanding the fate-determining nodes of bGPCs and provide valuable insights into the development and function of the VZ.

## TFEB activation restrains excessive multiciliogenesis

Strict control over the number and size of organelles is a prerequisite for maintaining cell homeostasis due to the limited availability of cellular components [78,79]. In particular, multiple motile cilia, a hair-like organelle protruding from the cell surface, require vast material consumption for their formation and substantial energy expenditure for beating [49,50]. Therefore, understanding the mechanisms that restrain excessive muliticilia generation is critical. We found that, during the differentiation of bGPCs into multiciliated EPCs, the expression of *Tfeb* was up-regulated (Fig 6A and 6B). Strikingly, *Tfeb* depletion facilitated EPC production, whereas its activation—via lysosomal, metabolic, or pharmacological perturbations—suppressed differentiation into EPC lineage. These findings suggest that the expression level of *Tfeb* is important for the proper differentiation of bGPCs. Under the normal developmental condition, the expression of *Tfeb* goes up in the EPC branch, acting as a "brake" to prevent excessive multiciliogenesis. Without *Tfeb*, bGPCs would predominantly differentiate into EPCs, and hence negatively impact the differentiation into NSC-NBs. Although the mechanistic data is based on *ex vivo* RGC culture, these findings provide valuable preliminary insights into the poorly understood mechanisms governing multiciliogenesis.

Furthermore, TFEB collaborates with other lineage-specific factors to enforce EPC fate restriction. For instance, TFEB partners with LHX2, a TF known to suppress multicilia formation, to directly repress genes critical for multiciliogenesis. Such partnerships illustrate how progenitors integrate metabolic and transcriptional signals to balance specialization with physiological constraints. By coupling lineage-specific repression with global homeostatic regulation, cells achieve precise control over energetically demanding processes like multiciliogenesis.

## Targeting *Tfeb* to tune postnatal NSC production could be a potential strategy for the prevention and treatment of NDDs

The major hallmarks of NDDs are accumulation of disease-associated proteins and extensive loss of neurons [80]. Recent studies have shown that *Tfeb* plays crucial roles in the NDD pathogenesis. Dysregulation of *Tfeb* expression, nuclear localization and transcriptional activity has been observed in various NDDs, including Alzheimer's Disease, Parkinson's disease and Huntington's disease [81]. The activation of *Tfeb* has been widely proven to ameliorate the pathological protein aggregates in neurons [66,81–83]. It promotes the removal of these aggregates through the up-regulation of lysosomal biogenesis and autophagy. Notably, the clinical drugs Aspirin and Rapamycin have shown promising results by enhancing TFEB activity and reducing protein aggregation in disease models [83–86].

Many preclinical studies and a few clinical trials have shown beneficial outcomes after NSC transplantation for NDDs [12,73]. In this study, we uncovered that Rapamycin administration during development partially rescued the spatial memory deficits of a NDD model. It implies that utilizing approved *Tfeb*-targeted agonists to modestly increase NSC production in the developing VZ may potentially compensate for neuronal loss in NDDs, leading to preventive or delaying effects on these conditions. This strategy holds promise because it circumvents several unresolved issues in the NSC transplantation approach, such as cell source selection and cell culture safety [12,73]. Further investigations are needed to fully understand the molecular mechanisms underlying *Tfeb*-mediated NSC production and to optimize the therapeutic strategy.

## Targeting *Tfeb* to regenerate EPCs could potentially contribute to hydrocephalus management

Hydrocephalus is a neurological disorder characterized by the abnormal CSF in the brain, resulting in increased intracranial pressure and ventricular enlargement [27]. The disorder can be classified as congenital hydrocephalus (primarily caused by genetic factor) or acquired hydrocephalus (secondary to infections, tumors, trauma, or hemorrhage) [27]. It has

been revealed that EPC detachment from the VZ correlates with altered CSF circulation and composition in hydrocephalic conditions [87]. Notably, a recent published paper demonstrated that GMNC or MCIDAS overexpression directs EPC-lineage reprogramming in animal models of hydrocephalus [88]. As *Tfeb* knockdown has been shown to promote EPC-lineage specification in our study, it is plausible to anticipate that AAV-mediated delivery of *Tfeb*-targeting shRNA could induce EPC regeneration in human hydrocephalus. The EPC restoration strategy could represent a significant advancement for hydrocephalus, offering a targeted alternative to neurosurgical shunting, the current standard treatment strategy with invasive procedures, complication risks and persistent neurocognitive impairments [89].

## Methods

### Ethics statement

C57BL/6J mice were maintained in laboratory animal center at Southern University of Science and Technology. Procedures were in accordance with the National Guideline for Ethic Review of Animal Welfare (GB/T 35892-2018) in China and approved by Experimental Animal Welfare Ethics Committee, Southern University of Science and Technology (SUSTech-JY2020211).

### Antibodies and oligonucleotide sequences

Antibodies used for immunoblotting, immunofluorescence or immunoprecipitation can be found in S1 Table. Sequences of siRNA, plate-based scRNA-seq and qPCR primers can be found in S2–S4 Tables.

### CD133-positive RGC isolation by FACS

Dissection of VZs from neonatal mouse brains was performed as described [58]. Briefly, the midbrain, the olfactory bulb, meninges and the hippocampus were removed in sequence to attain the telencephalon. Then VZ-containing tissues were separated from the telencephalon. Tissues from 4 mice were pooled, transferred to a 1.5-mL Eppendorf tube, and digested with freshly prepared Papain solution (10 U/mL, Worthington, cat. no. LS003126) at 37 °C for 30 min. Then the Papain solution was aspirated, and 1 mL culture medium, which consisted of DMEM (Gibco, cat. no. SH30256.01) supplemented with 20% FBS (Gibco, cat. no. 30044333), 100 U/ml penicillin and 100 µg/mL streptomycin sulfate (Hyclone, cat. no. SV30010), was added and incubated for 1 min to terminate the digestion. The supernatant was removed, and another 1 mL culture medium (20% FBS) was added. The digested tissues were gently pipetted ups and downs 20 times using a P1000 tip to dissociate the cells, and then cells were spun down in a centrifuge at 350 RCF for 5 min. The supernatant was discarded, and 1 mL blocking buffer (1% BSA, Sigma, cat. no. V900933 in PBS, Hyclone, cat. no. SH30256.01) was used to resuspend the cell pellet. Cells were spun down again at 350 RCF for 5 min and the supernatant was discarded. A 200 µL blocking buffer containing 2 µL CD133-PE antibody (eBioscience, cat. no. 12-1331-82) or the IgG isotype control (eBioscience, cat. no. 12-4301-82) was added to resuspend the cell pellet and label the cells on ice for 30 min. Cells were then washed once with 1 mL blocking buffer and finally resuspended in 1 mL blocking buffer containing 1 µg/mL DAPI (Sigma, cat. no. D8417). The cell suspension was transferred to a FACS tube for sorting. CD133-positive, DAPI-negative single cells were collected into a 15 mL tube or sorted into each well of 384-well plates for scRNA-seq library construction.

### Single-cell RNA-seq library construction

For the droplet-based scRNA-seq experiment, two biological replicates each of RGCs from P0 and P5 were used. The V3 chemistry of the 3′ Single Cell Gene Expression kit was used according to the 10× Genomics user guide CG000204 Rev D. For the plate-based scRNA-seq experiments, two replicates of RGCs from P0 were used. The experiments were performed exactly according to the step-by-step protocol described previously [36], where the cDNA was amplified for 11 cycles, and the final library was amplified for 11 cycles.

## Single-cell RNA-seq data analysis

For scRNA-seq data, reads were processed using the STARsolo [90] pipeline as previously described [36]. FastQ files generated using the Illumina NovaSeq 6000 were aligned to the mm10 mouse reference genome, and a gene expression matrix containing the UMI counts for each gene in each cell was obtained. This output was imported into the R toolkits for downstream analyses. Cell doublets in 10× data were identified and filtered using the DoubletFinder package [91]. The endothelial cells, microglia and pericytes/fibroblasts, which were annotated by the SingleR package [92], were not included in the analysis. Dying cells, which showed low number of detected genes and UMIs and high mitochondrial genes were also excluded from the subsequent analyses. The remaining cells from different biological replicates in the 10× data or from different plates in the plate data were merged using Seurat [93]. The top 2000 highly variable genes were obtained and used for the principal component analysis (PCA). Harmony algorithm [37] was applied to remove batch effect among different biological replicates. The first 30 Harmony dimensions were used for graph-based clustering to identify distinct groups of cells. These groups were projected onto 2D tSNE plane. Then dimensionality reduction was done through the DDRTree method and the trajectory of cellular differentiation was constructed using the 'orderCells' function. Pseudotime values were acquired by setting the bGPC state as the root. Differential expression analyses to identify the cluster markers and the branch markers were performed using the 'FindAllMarkers' function in Seurat and the 'BEAM' function in Monocle2, respectively. Trajectory analysis using the diffusion map algorithm was completed with the destiny package [39]. GO analysis for enriched biological processes was performed using clusterProfiler package [94]. The exact code and parameters of the procedures can be found in the GitHub repository: https://github.com/sibszheng/VZ_development. The gene list used for cell clustering is provided in S5 Table.

## In vitro culture of EPCs

In vitro cultured EPCs were obtained as described [58], with some modifications. The VZs from 4 P0 mice were pooled, transferred to a 1.5 mL Eppendorf tube, and digested with freshly prepared Papain solution at 37 °C for 30 min. Then the 10 U/mL Papain solution was aspirated, and 1 mL culture medium (20% FBS) was added, followed by 1 min of incubation to terminate the digestion. The supernatant was removed, and another 1 ml culture medium (20% FBS) was added. The digested tissues were gently pipetted up and down by 20 times with a P1000 tip to dissociate the cells mechanically. A 25 cm$^2$ flask was pre-coated with 5 µg/mL Laminin (ThermoFisher, cat. no. 23017015) in PBS for 8–12 h, rinsed twice with PBS, and then 3 mL culture medium (20% FBS) was added. The cells were seeded into the flask and cultured at 37 °C in an atmosphere containing 5% $CO_2$. After 24 h, the old medium was replaced by fresh culture medium (20% FBS). After another 24 h, the old medium was aspirated, and the flask was vigorously shaken to remove the neuroblasts and neurons. The remaining radial glia-enriched cells were rinsed once with PBS and further cultured to 30%–40% confluency. Then the culture medium was aspirated, washed once with PBS, and 1 mL of 0.05% Trypsin (Gibco, cat. no. 25300062) was added for digestion at 37 °C for 5 min. A 2 mL culture medium (20% FBS) was added to terminate the digestion. The cells were transferred to a 15 ml tube and centrifuged at 900 RPM for 5 min. The supernatant was discarded, and 1 mL culture medium (20% FBS) was added to resuspend the cell pellet. A 29-mm glass-bottom dishes (Cellvis, cat. no. D29-14-1.5-N) were pre-coated with 5 µg/mL Laminin in PBS for 8–12 h and rinsed twice with PBS before use. 250 µL of cell suspension was transferred into each well of the dish and incubated at 37 °C until the cells were completely adhered. After adding 1 mL culture medium (10% FBS), the cells were further cultured to 100% confluency, and then maintained in starvation medium (culture medium without FBS) to induce differentiation into EPCs.

## Plasmids

The full-length *Tfeb* (NM_001161722), *Lhx2* (NM_010710), *Gmnc* (NM_001013761), and *Gmnn* (NM_020567) were amplified by PCR from total cDNAs of mouse EPCs and constructed into pLV-EGFP-C1 or pcDNA3.1-FLAG to express

GFP-tagged or FLAG-tagged fusion proteins, respectively. The cDNAs for the S121A mutant (TCC→GCC), the S141A mutant (AGT→GCT), the S210A mutant (TCC→GCC), the S466A mutant (AGC→GCC), the ΔNLS mutant (1,032 AGAAGACGCAGG→GCAGCAGCCGCG), and the RNAi-resistant constructs of *Tfeb* (956 GCGAGAGCTAACAGAT-GCT→AAGGGAATTGACTGACGCA) were produced by PCR. All the constructs were verified by sequencing.

## siRNA and plasmid transfection

For centriole staining, siRNAs were transfected into cultured RGCs using Lipofectamine RNAiMAX (ThermoFisher, cat. no. 13778150) at serum starvation day −1 (SS d − 1). For one 29-mm dish, the original medium was discarded and replaced by 500 µL fresh culture medium (10% FBS) or starvation medium before transfection. 2 µL of 20 µM siRNA (Genepharma) were mixed in 125 µL Opti-MEM (Gibco, cat. no. 31985070) by vortexing. A 3 µL Lipofectamine RNAiMAX were mixed in 125 µL Opti-MEM by vortexing and incubated at room temperature for 5 min. The two diluents were then mixed by vortexing and incubated at room temperature for 20 min. Finally, the mixture complex was added to the culture medium (10% FBS) or starvation medium. After 24 h of transfection, cells were rinsed with PBS twice and serum-starved for centriole staining at SS d + 2. For cilium staining, additional siRNA transfection was performed at SS d + 2 and cells were harvested at SS d + 5.

For the expression of GFP-tagged proteins, cultured RGCs were transfected using Lipofectamine 2000 (ThermoFisher, cat. no. 11668019) at SS d − 1. For one 29-mm dish, the original medium was discarded and replaced by 500 µL fresh culture medium (10% FBS) before transfection. 2 µg plasmid were mixed in 125 µL Opti-MEM by vortexing. 1.5 µL Lipofectamine 2000 were mixed in 125 µL Opti-MEM by vortexing and incubated at room temperature for 5 min. Then the two diluents were mixed together by vortexing and incubated at room temperature for 20 min. Finally, the mixture complex was added to the culture medium. After 24 h of transfection, cells were rinsed with PBS twice, subjected to serum starvation to observe EPC-lineage differentiation at SS d + 5 or maintained in culture medium (10% FBS) for NSC-lineage differentiation assay.

For the rescue experiments, cultured RGCs were transfected using Lipofectamine 2000 at SS d − 1 to express RNAi-insensitive GFP-TFEB and GFP on top of the transfection of siRNAs. For the co-immunoprecipitation experiments, HEK293T cells were transfected using Lipofectamine 2000 to express the exogenous proteins.

## Drug treatment

Cultured RGCs were incubated with 1–100 mM trehalose (Selleck, cat. no. S3992), 100 mM sucrose (Sigma, cat. no. V900116), 250 nM Torin1 (Selleck, cat. no. S2827), 4 µM cyclosporin A (Selleck, cat. no. S2286) or 5 µM Rapamycin (MCE, cat. no. HY-10219) in starvation medium from SS d0 to SS d + 2. For amino acid starvations, cells were cultured in DMEM without amino acids (Wako, cat. no. 048-33575) from SS d0 to SS d + 2. Cells were fixed and subjected to immunostaining after drug treatment.

## Immunofluorescent staining of cultured cells

Cells were fixed with 4% paraformaldehyde (PFA) in PBS for 15 min at room temperature and permeabilized with 0.5% Triton X-100 for 15 min. For centriole staining, cells were pre-extracted with 0.1% Triton X-100 for 30 s before fixation. After 1-h blocking at the room temperature with the blocking buffer (1% BSA in PBS), the samples were labelled with primary antibodies (diluted in the blocking buffer) and incubated overnight at 4 °C. Subsequently, the samples were rinsed three times with the blocking buffer, followed by incubation with secondary antibodies and 1 µg/mL DAPI (diluted in the blocking buffer) for 1 h at the room temperature. The samples were rinsed three times with PBS and then mounted using an anti-fade mounting medium (Dako, cat. no. s3023). Multi-layered confocal images were captured by using Leica TCS SP8 system and processed with maximum intensity projections. Cells possessing more than four centrioles (γ-TUB) or

multiple cilia (ARL13B) were classified as EPCs. For Figs 5–7 and S7–S9 Figs, at least 151 cells from 6 images were quantified in each experiment and condition. For Fig 8, at least 64 GFP-positive cells from 43 images were quantified in each experiment and condition. Two-sided Student *t* test was used to calculate *P*-values between unpaired samples.

## Tissue sectioning and immunostaining

Cryo-sectioning was employed for 20-μm-thick tissue sections. For P0 mice, brains were dissected and fixed in 4% PFA at 4 °C for 4 h. For P5 and P15 mice, perfusion with PBS followed by 4% PFA was conducted prior to the dissection. The fixed brains were soaked overnight in 30% sucrose at 4 °C for dehydration. Next, the brains were embedded in OCT compound (Leica, cat. no. 14020108926) and coronally cryo-sectioned using a Leica CM1950 cryostat microtome. The sections were collected onto glass slides.

To perform immunostaining, the cryo-sections were first rinsed with PBS to eliminate the OCT compound. Subsequently, the sections were incubated in blocking buffer (10% normal goat serum and 0.3% Triton X-100 in PBS) for 1 h at room temperature to prevent non-specific binding. Next, the sections were labeled with primary antibodies in the blocking buffer overnight at 4 °C. After three rinses with the blocking buffer, the samples were incubated with secondary antibodies and 1 μg/mL DAPI in the blocking buffer for 1 h at room temperature. The sections were then rinsed three times with PBS and mounted in anti-fade mounting medium. For Fig 2F, at least 164 cells from 6 images of P0 brain sections were quantified in each experiment.

## ChIP-seq

ChIP-seq experiments were performed using the ChIPmentation protocol as previously described [95]. Briefly, $5 \times 10^6$ EPCs at SS d + 5 were collected and crosslinked with 1% formaldehyde in PBS. Then the crosslinking was stopped by adding glycine to a final concentration of 125 mM. The cells were washed twice with PBS and resuspended in 300 μL Sonication/IP buffer. The chromatin was fragmented by sonication using a Bioruptor Pico for 4 min (30 s on, 30 s off). The sonicated chromatin was centrifuged at 16,000 RCF for 10 min at 4 °C. The supernatant was incubated with 10 μL protein A Dynabeads (ThermoFisher, cat. no. 10001D) pre-bound with 1 μg anti-TFEB antibody (Bethyl Laboratories, cat. no. A303-673A) on a rotator overnight in a cold room. Then the IP was washed once with RIPA Wash Buffer, once with Low Salt Wash Buffer, once with High Salt Wash Buffer, once with LiCl Wash Buffer and twice with 10 mM Tris-HCl (pH 8.0). The beads were resuspended in 30 μL tagmentation mix containing 1 μL Tn5 and incubated on a thermomixer at 37 °C for 5 min. Finally, the beads were washed twice with the low salt buffer and once with TE. Subsequently, the beads were resuspended in 100 μL ChIP elution buffer and incubated at 65 °C overnight for the reverse crosslinking. The tagmented DNA was purified using the ZYMO DNA Clean and Concentrator-5 kit. The library was prepared by using standard Nextera PCR primers.

## Bulk RNA-seq library construction

At SS d + 5, total RNA of EPCs treated with NC or *Tfeb-si1* were extracted using the RNAsimple Total RNA Kit (TIANGEN, cat. no. DP419). Library construction was performed using the SHERRY protocol as described previously [96]. Briefly, the amount of RNA was quantified using a Nanodrop. A 2 μg total RNA were mixed with 2 U Rnase-free Dnase (Promega, cat. no. M6101) and incubated at 37 °C for 30 min. The RNA was purified by 2× VAHTS DNA Clean beads (Vazyme, cat. no. N411). Then 200 ng purified RNA was used for reverse transcription using 100 U Maxima H Minus Reverse Transcriptase (ThermoFisher, cat. no. EP0752). The resulting RNA/cDNA hybrid was tagmented with 2.5 μL Tn5. The tagmented product was purified using 2× VAHTS DNA Clean beads. Library preparation was done using standard Nextera primers.

## ChIP-seq and bulk RNA-seq data analysis

For ChIP-seq data, reads were mapped to the mm10 mouse reference genome using hisat2 [97]. Then the reads with mapping quality less than 30 were removed by samtools [98] and deduplicated using the 'MarkDuplicates' function from

the Picard tool (https://broadinstitute.github.io/picard/). Peaks were called on the output BAM file using MACS2 [99]. BedGraph files generated from MACS2 callpeak were converted to bigWig files and visualized via UCSC genome browser [100]. Three biological replicates were performed. Only peaks that were present in both biological replicates were considered. De novo Motif discovery was performed using findMotifsGenome.pl in the HOMER suite [101] on the narrowPeak files returned from MACS2.

For bulk RNA-seq data, reads were mapped to the mm10 mouse reference genome using hisat2 and supplied with gene annotation from GENCODE vM25 [102]. Gene expression was quantified by HTSeq [103] and differential gene expression analysis was conducted using DESeq2 [104] package. Three biological replicates for *NC* or *Tfeb-si1* treated group were performed. The criteria for the identification of differentially expressed genes were adjusted $P$ value < 0.001 and |$\log_2$fold change| > 0.5. Multiple testings were corrected by calculating false discovery rate (FDR).

The exact code and parameters of the procedures can be found in the GitHub repository: https://github.com/sibszheng/VZ_development.

## Immunoblotting

Cultured cells were directly lysed in 1× SDS-PAGE loading buffer (Biosharp, cat. No. BL502B) and boiled at 99 °C for 10 min. Proteins separated by SDS-PAGE (Sangon, cat. no. C681102) were transferred to nitrocellulose membranes. Blots were blocked with 3% BSA diluted in TBS (Sangon, cat. no. B548105) with 0.05% Tween-20 (TBST) for 1 h at room temperature and then incubated with primary antibodies (diluted in 1% BSA in TBST) at 4 °C overnight. After extensive rinse with TBST, membranes were incubated with secondary antibodies (diluted in 1% BSA in TBST) at room temperature for 1 h. After thorough wash in TBST, protein bands were visualized with enhanced chemiluminescent reagent (Bio-Rad, cat. no. 1705061) and exposed to Tanon 5200 Chemiluminescent Imaging System.

## Immunoprecipitation

Cells were lysed in Sonication/IP Buffer containing protease inhibitor cocktail (1:200, Abcam, cat. no. ab201111) and phosphatase inhibitor cocktail (1:100) (Sigma, cat. no. P0044). Without fixation and sonication, the lysate was centrifuge at 16,000 RCF for 10 min at 4 °C, and the supernatant was incubated with Protein A Dynabeads pre-bound with TFEB antibody or Pan Mouse IgG Dynabeads (ThermoFisher, cat. no. 11041) pre-bound with FLAG antibody overnight at 4 °C. After beads were washed once with RIPA Wash Buffer, once with Low Salt Wash Buffer, once with High Salt Wash Buffer, once with LiCl Wash Buffer and twice with 10 mM Tris-HCl (pH 8.0), proteins associated with the beads were eluted in 1× SDS-PAGE loading buffer at 99 °C for 10 min. A $5 \times 10^6$ cells and 1 µg antibody were used per IP.

## Rapamycin administration and MWM

Wildtype mice and 5 × FAD mice were purchased from Cyagen Biosciences. Breeder pairs were established and monitored daily. Male mice were removed from cages upon visual confirmation of pregnancy in females. Dams with newborns were provided ad libitum access to chow containing encapsulated Rapamycin (42 mg/kg). Pups were separated from dams and maintained on Rapamycin until 3 months of age. Then spatial learning and memory were assessed using the MWM. The procedure of MWM was described before [105] with a little modification. Trials were recorded and analyzed with automated video-tracking software (ANY-maze, Stoelting Co., USA). The MWM consisted of a circular tank (120 cm in diameter), filled with opaque water (20 ± 1 °C). On the rim of the tank were four marks that were 90 degrees apart (North – N, South – S, East – E, West – W). Thus, the maze was divided into four quadrants. An escape platform (10 cm in diameter) was placed in the pool at a specified location. In the visible platform trail, the top of the platform was 0.2 cm above the water surface and a flag painted black was placed on the center of the platform to increase its visibility. Mice were tested with 4 trials. The starting position from which the mice were placed into the maze and position of the platform

was changed in each trial. The mice were allowed to search for the visible platform for 60 s. If a mouse failed to get on the platform within 60 s, it was guided to the platform and allowed to rest for 15 s. The average moving speed and the time taken to get on the platform were recorded. In the hidden platform training phase, the platform without flag decoration was placed in the center of southwest quadrant and submerged 0.5 cm below the water surface. Mice were trained for 5 days with 4 trials/day to locate the hidden platform. The position of the platform was the same in all trials. Each trail the mice were placed into the maze at 1 of 4 random points and allowed to search for the hidden platform for 60 s. If a mouse failed to find the platform within 60 s, it was guided to the platform and allowed to rest for 10 s. The time taken to find the hidden platform was recorded. During the probe test, the platform was removed from the pool, and the mouse was allowed to swim freely for 1 min. The number of crossings over the platform position and the time spent in target quadrant were scored.

## Supporting information

**S1 Fig. Quality control, trajectory inference and marker gene expression of scRNA-seq. (A)** Distribution and sorting gates of CD133-labeled cells. IgG served as a negative control for defining the sorting region (indicated by the rectangle in the bottom plot). **(B)** UMI count, gene count, and mitochondrial gene percentage for each individual cell in each replicate of 10× and the Plate data. **(C)** 2D t-SNE visualization of 30,445 cells from the 10× data (left) and 2,594 cells from the Plate data (right). Individual cells are color-coded according to replicates. The replicates of 10× data were integrated using Harmony algorithm. **(D and E)** 2D t-SNE visualization **(D)** of the Plate data and the bifurcating trajectory **(E)** constructed by Monocle. Individual cells are color-coded according to clusters or states. **(F)** The bifurcating trajectory inferred using the diffusion map algorithm. **(G)** Expression profiles of additional markers used to assign cell classifications of bGPCs, nEPCs and nNSC-NBs. **(H)** Expression dynamics of various marker genes over pseudotime. (PDF)

**S2 Fig. GO analysis of the nNSC-NB branch and the validation of OPC-NB bipotent precursors. (A)** Top 12 GO terms of biological processes from genes differentially expressed in the four clusters along the nNSC-NB branch. **(B)** Expression profiles of six well-known OPC marker genes along the bifurcating trajectory. **(C)** Immunofluorescence analyses of neonatal brain sections showing the co-expression of the NB marker (TUBB3, DCX) and the OPC marker (PDGFRA or NKX2-2). Arrows indicate double-positive cells. LV, lateral ventricle. The scale bar is 20 μm. (PDF)

**S3 Fig. Multiciliogenesis program dominates bGPC transition into nEPCs. (A)** Top 10 GO terms of biological processes from genes differentially expressed in the two clusters along the nEPC branch. **(B)** Heatmap showing the dynamic expression of multicilia-specific genes along the nEPC branch. **(C)** The four multicilia-specific genes absent in our data were also not detectable in the adult VZ scRNA-seq and ependymal cilia proteomic data. (PDF)

**S4 Fig. Analysis of EPC-fate specific genes. (A)** Heatmap showing the dynamic expression of the top 3,000 branch-specific differentially expressed genes. **(B)** Top 10 GO terms of biological processes enriched in 1,172 EPC-fate specific genes. **(C)** The Venn diagram illustrating that 303 EPC-fate specific genes are included in the cilium gene set and the table showing the high enrichment of cilium genes in the EPC-fate specific genes. **(D)** Top Go terms of biological processes enriched in 869 non-ciliary EPC-fate specific genes. **(E)** Expression profiles of hydrocephalus genes and EPC-fate regulators in the bifurcating trajectory. **(F)** Quantification results for Fig 4G. Three independent experiments were performed. Error bars represent SD. Asterisks indicate $P$-values from Student's $t$-tests, *$P < 0.05$; **$P < 0.01$; ***$P < 0.001$; ns, not significant. The data underlying this figure can be found at S1 Data, specifically in the sheet labeled 'S4 Fig'. (PDF)

**S5 Fig.** *Npas1* and *Foxa2* are more abundant in EPCs. **(A and C)** Expression profiles of *Npas1* **(A)** and *Foxa2* **(C)** along the bifurcating trajectory. **(B and D)** Expression values of *Npas1* **(B)** and *Foxa2* **(D)** in the indicated cell types of the developing mouse brain dataset. **(E)** Immunoblotting showed efficient depletion of *Foxj1* in EPCs by RNAi. siRNA transfection was performed at SS d − 1 and SS d + 2 then cells were harvested at SS d + 5. The quantification results were from three independent experiments. Error bars represent SD. Asterisks indicate *P*-values determined by Student's *t*-tests between NC and siRNA-treated groups, ***$P < 0.001$. The data underlying this figure can be found at S1 Data, specifically in the sheet labeled 'S5 Fig'.
(PDF)

**S6 Fig.** Lysosomal gene expressions during EPC-lineage differentiation are independent of *Tfeb*. **(A)** Expression profile of *Tfeb* along the bifurcating trajectory. **(B)** Expression values of *Tfeb* in the indicated cell types of the developing mouse brain dataset. **(C)** Heatmap shows the dynamic expression of lysosome genes with CLEAR motif along the nEPC branch. **(D)** Principal component analysis of three replicates of non-targeting control and *Tfeb-si1* samples from the bulk RNA-seq experiments.
(PDF)

**S7 Fig.** TFEB functions downstream of GMNC during EPC-lineage differentiation. **(A)** QPCR showed efficient depletions of *Gmnc* and *Gmnn* in EPCs. siRNA transfection was performed at SS d − 1 then cells were harvested at SS d + 2. **(B)** Knockdown of *Gmnc* and *Gmnn* inhibited and promoted the EPC generation, respectively. Centrioles were labelled by γ-TUB, and cell borders were labelled by β-CAT. At least 261 cells were quantified in each experiment and condition. **(C)** QPCR showed deficiency of *Gmnc* but not *Gmnn* reduced *Tfeb* expression. **(D)** Expression levels of *Gmnc* and *Gmnn* from bulk RNA-seq results. All of the quantification results above were from three independent experiments. Error bars represent the SD. Asterisks indicate *P*-values determined by Student's *t*-tests between NC and siRNA-treated groups, *$P < 0.05$; ***$P < 0.001$; ns, not significant. **(E)** Co-IP assays revealed the absence of direct physical interactions between TFEB and either Geminin protein in HEK293T cells. GFP was used as control. FLAG-GMNC or FLAG-GMNN combined with GFP or GFP-TFEB were expressed in HEK293T cells for 36 h and then harvested for FLAG IP. Input samples, 10% and IP samples, 50% were subjected for immunoblotting analysis. The data underlying this figure can be found at S1 Data, specifically in the sheet labeled 'S7 Fig'.
(PDF)

**S8 Fig.** Activation of TFEB blocks the GPC to EPC differentiation. **(A)** TFEB translocated from cytosol to nucleus after 100 mM sucrose treatment. At least 151 cells were quantified in each experiment and condition. **(B)** A dramatic reduction in the EPC percentage after 100 mM sucrose treatment. At least 233 cells were scored in each experiment and condition. **(C)** Amino acid starvation induced translocation of TFEB from the cytosol to the nucleus. At least 151 cells were quantified in each experiment and condition. **(D)** EPC-lineage differentiation was drastically suppressed after amino acid starvation. At least 233 cells were scored in each experiment and condition. All of the quantification results above were from three independent experiments. Error bars represent the SD. Asterisks indicate *P*-values from Student's *t*-tests, ***$P < 0.001$. The scale bars above are 20 μm. **(E)** UCSC genome browser track showing the good quality of the TFEB ChIP-seq data. **(F)** Among the four reported consecutive active mutants of TFEB, only TFEB-S210A showed predominant nuclear localization when transfected into GPCs. The scale bar is 25 μm. The data underlying this figure can be found at S1 Data, specifically in the sheet labeled 'S8 Fig'.
(PDF)

**S9 Fig.** Rapamycin administration during development mitigates memory deficits of 5× FAD mice. **(A)** Immunofluorescence analyses showing the translocation of TFEB from the cytosol to the nucleus upon 5 μM Rapamycin treatment. The quantification results were from three independent experiments. At least 288 cells were scored in each experiment

and condition. The scale bar is 20 μm. **(B)** Quantification of moving speed of 3-month-old mice in each group during visible platform trail. **(C)** Quantification of escape latencies during visible platform trail. **(D)** Quantification of escape latencies during 5-day hidden platform trail. **(E)** Quantification of platform crossover number during probe trial. **(F)** Average movement heatmap of mice in each group during probe trial. The circle indicates the platform position. 5–9 mice were used for each group. Error bars represent SD. Asterisks indicate *P*-values from Student's *t*-tests, *$P < 0.05$; **$P < 0.01$; ns, not significant. The data underlying this figure can be found at S1 Data, specifically in the sheet labeled 'S9 Fig'.
(PDF)

**S1 Table.  List of antibodies used for immunoblotting, immunofluorescence or immunoprecipitation.**
(DOCX)

**S2 Table.  List of siRNA sequences.**
(DOCX)

**S3 Table.  List of plate-based scRNA-seq primer sequences.**
(DOCX)

**S4 Table.  List of qPCR primer sequences.**
(DOCX)

**S5 Table.  List of top 2,000 highly variable genes for cell clustering in Fig 1D and 1G.**
(CSV)

**S1 Raw Images.  Original blot images for immunoblotting data.**
(PDF)

**S1 Data.  Source data for main figures and supplementary figures.**
(XLSX)

## Acknowledgments

We thank all members from the Chen and Song labs for the helpful discussion of the project. We thank Xibin Lu for the excellent support of FACS. We acknowledge the assistance of SUSTech Core Research Facilities. The computational work was supported by Center for Computational Science and Engineering at Southern University of Science and Technology.

## Author contributions

**Conceptualization:** Jianqun Zheng, Xi Chen.

**Data curation:** Jianqun Zheng, Yawen Chen, Yujian Zhu.

**Formal analysis:** Jianqun Zheng, Yukun Hu, Jie Lin, Yunlong Zhang.

**Funding acquisition:** Xi Chen.

**Investigation:** Jianqun Zheng, Yawen Chen, Yukun Hu, Yujian Zhu, Jie Lin, Manlin Xu.

**Methodology:** Jianqun Zheng, Yawen Chen, Xi Chen.

**Project administration:** Jianqun Zheng, Weihong Song, Xi Chen.

**Resources:** Weihong Song, Xi Chen.

**Supervision:** Weihong Song, Xi Chen.

**Validation:** Jianqun Zheng, Yawen Chen, Yukun Hu, Yujian Zhu, Manlin Xu.

**Visualization:** Jianqun Zheng, Yawen Chen, Xi Chen.

**Writing – original draft:** Jianqun Zheng, Weihong Song, Xi Chen.

**Writing – review & editing:** Jianqun Zheng, Yawen Chen, Weihong Song, Xi Chen.

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
