## [Editor Report · Decision Letter 0]

Dear Dr Zheng, 

Thank you for submitting your manuscript entitled "Unraveling Lineage Roadmaps and Fate Determinants to Postnatal Neural Stem Cells and Ependymal Cells in the Developing Ventricular Zone" for consideration as a Research Article by PLOS Biology.

Your manuscript has now been evaluated by the PLOS Biology editorial staff as well as by an academic editor with relevant expertise and I am writing to let you know that we would like to send your submission out for external peer review.

Once your full submission is complete, your paper will undergo a series of checks in preparation for peer review. After your manuscript has passed the checks it will be sent out for review. To provide the metadata for your submission, please Login to Editorial Manager (https://www.editorialmanager.com/pbiology) within two working days, i.e. by Aug 31 2024 11:59PM.

Kind regards,

Suzanne

Suzanne De Bruijn, PhD, 

Associate Editor

PLOS Biology

sbruijn@plos.org

---

## [Decision Letter · Decision Letter 1]

Dear Dr Zheng,

Thank you for your patience while your manuscript "Unraveling Lineage Roadmaps and Fate Determinants to Postnatal Neural Stem Cells and Ependymal Cells in the Developing Ventricular Zone" was peer-reviewed at PLOS Biology. Your manuscript has been evaluated by the PLOS Biology editors, an Academic Editor with relevant expertise, and by several independent reviewers.

As you will see in the reviewer reports, which can be found at the end of this email, although the reviewers find the work potentially interesting, they have also raised a substantial number of important concerns. Based on their specific comments and following discussion with the Academic Editor, it is clear that a substantial amount of work would be required to meet the criteria for publication in PLOS Biology. However, given our and the reviewer interest in your study, we would be open to inviting a comprehensive revision of the study that thoroughly addresses all the reviewers' comments. Given the extent of revision that would be needed, we cannot make a decision about publication until we have seen the revised manuscript and your response to the reviewers' comments. Your revised manuscript would need to be seen by the reviewers again, but please note that we would not engage them unless their main concerns have been addressed. 

You will see that both reviewers find the conclusions potentially interesting, but they also raise serious concerns regarding the methodology used in the study. Reviewer 1 is not convinced about the use of CD133 as a selection marker because it is known to be problematic. Reviewer 2 questions the lack of replicates for scRNA-seq and bulk RNA-seq experiments, among other issues.

We appreciate that these requests represent a great deal of extra work, and we are willing to relax our standard revision time to allow you 6 months to revise your study. Please email us (plosbiology@plos.org) if you have any questions or concerns, or envision needing a (short) extension.

**IMPORTANT - SUBMITTING YOUR REVISION**

*Resubmission Checklist*

*Published Peer Review*

*PLOS Data Policy*

*Blot and Gel Data Policy*

Sincerely,

Suzanne

Suzanne De Bruijn, PhD, 

Associate Editor

PLOS Biology

sbruijn@plos.org

REVIEWS:

Reviewer #1: 

The paper has interesting findings summarised below:

- There are three distinct cellular states in the developing ventricular zone forming a continuous bifurcating trajectory (bGPCs, nNSCs-NBs, n-EPCs).

- A novel intermediate state of cells expressing markers of oligodendrocytes and neuroblasts was detected.

- Transcriptional factors essential for ependymal cell lineage differentiation were identified.

- TFEΒ was pointed out as a master regulator, tuning neural stem cell and ependymal cell bifurcation, in cooperation with LHX2.

The authors unravel parts of the molecular mechanism underlying cell fate decisions of progenitor cells towards neural stem or ependymal cell fates. As the common origin of NSCs and ECs have been previously described and the involvement of Geminin family members in this decision it would be required the authors to show which is the cooperation of TFEB with the pathway regulated by Geminin and GemC1.

Introduction details of the already known molecular pathway regulating cell fate decisions of ependymal cells should be added as there are used later in the text.

CD133 is also expressed in neural stem cells and in mature ependymal cells constituting a problem in previous studies when trying to discriminate neural stem cells and ependymal cells. How authors have verified that CD133 is specifically expressed in radial cells at that time point you're performing the experiments and you are not including populations of neural stem or ependymal cells in your analysis?

Percentages from quantifications are not provided in all graphs and should be added.

Discussion needs to be more general and not so focused on the figures that you have analysed before in the results section. 

Please check that you refer to the right figures when trying to support your statements.

In the section describing immunofluorescent staining of cultured cells/ tissue sectioning and immuno-staining/ immunoblotting / immunoprecipitation, antibodies/antibody-bound beads calalogue numbers and used dilutions are not mentioned and should be included. Issue the way that qualifications were done should be explained (How many fields and cells were counted and how statistics were performed).

The immunofluorescence figures are of poor quality the positive cells cannot be discriminated from the background.

The ependymal culture protocol in which you are referring never obtains neuronal cells can be specify whether you have made any alterations to the protocol. 

Minor comments 

line 69: instead of rhythmic beating use the term coordinated

line 73: disruption of the circulation of CSF is one of the causes that can possibly lead to hydrocephalus development- there are also other reasons that could results to hydrocephalus

line 233-236: it is proposed that ciliary genes may be implicated in bGPC-nNSC differentiation. Do all bGPCs express ciliary markers and later on there is mechanism that regulates their cell fate towards NSCs or ECs? Or this is a distinct subpopulation of bGPCs from the one you mentioned before expressing qNSC markers ?

line 251: use hydrocephalus development than formation

line 256: beating of cilia is the main propelling force for CSF circulation in mice. So it should be clarified because this does not apply for humans

line 271: the word Six- I guess you mean Six3

line 299: You refer to figure 4F but this is not the right figure

line 313-314: It is mentioned that "loss of LhX2 promoted EPC-lineage specification (fig 5A,B) which agreed with the in vivo results reported later".

In which in vivo experiments do you refer to?

In fig.4E is mentioned that reduction of LhX2 is related to NSC formation, - we would expect ependymal cells to be reduced and have an increase in neural stem cells in fig 5 as well.

line 333: it is mentioned that Foxa2 depletion did not affect centriole amplification in early stages of EPC lineage differentiation but affected cilium growth, resembling Foxj1 deficiency. In figure 5H only cilia are shown and it's not clear what happens with Foxj1 regarding centriole amplification -it would be nice to have a clear comment and the comparison between the two molecules.

line 366: Hes1 decrease is not obvious in the fig.6G. Is it statistically important?

line 451: just a typo of in deed - the space should be deleted

line 1237: the word facilitate gives a very strong statement- maybe just mention that play role in EPC lineage differentiation

line 1238: why different methodologies were used for McIdas and Lhx2?

line 1240: promoted and inhibited are mentioned in reverse order according to your previous results

line 1256: the time points that the experiment was performed should be included

line 1258 , 1272 and 1280: add the b-catenin marker used as cell border marker

line 1259 and 1282: identify the way you quantified the ECs- probably through g-tub accumulation?

line 1269: Activated TFEB _delete the word of

line 1275: quantifications of Fig.7C should be added

line 1285: Fig. 7G should be explained further, including more details for the experiment

line 1289: identify the details for the experiment (in which time points and and the conditions that the si RNA was performed)

line 1295: Maybe the statement that they are neuron-like cells should be done later when you use markers that show that.

line 1299 and 1302: Why different conditions are used in the two experiments (5 days serum starvation/ presence of serum)? 

Fig. 1 : for clustering in C some genes are shown in D. Based on which genes the clustering in E was done? A list should be provided.

Fig.2: in line 29 you mention that bGPGs can give rise to nNSCs-NBs and nEPCs. So maybe in the illustration 2B bGPGs should have a neutral colour or some EPCs should be yellow in order to show that they derive from bGPGs.

Fig.8 : the markers should be appeared in all images in order to be easier to follow and maybe the order of figures should be changed and show they graphical abstract of your results as a final point.

Are there any qualifications for these experiments in order to do show how strong your result is? 

line 473: the term primary niche EPCs is not clear - do you mean neonatal as mentioned in the text above?

Line 511: you mention that you identified transcriptional factors that play it all in fate determination towards EPCs or NSC-NBS and refer to figure 4C that contains data only for EPCs.

Line 529: you mention that Tfeb was upregulated both in vivo and in vitro, refering to figure 6A and B- could you please identify the experiments that were done in vivo?

Line 572: I would suggest making also a hypothesis for possible treatments of hydrocephalus using this knowledge as you have stated in the abstract

Line 585: add in the title that this is the staining protocol for FACS 

Line 593, 658 : you mention that culture medium contains 20% FBS. Later on in the protocol for example in line 673 you use 10% FBS. Why is this different percentage of FBS used? Have you characterised the cell culture of EPCs? If yes which is the percentage of ependymal cells when culturing under the conditions you mentioned above?

Line 697: you mention that RNAi Was done in culture medium containing 10% FPS or starvation medium. How did you decide to do the procedure in starvation medium while differentiation has already started? It would be ideal to clarify in each experiment at which time point the RNAi has been performed in the corresponding figure.

Reviewer #2: This manuscript presented a single cell study of the developing ventricular zone (VZ) from mice and identified three distinct cellular states of radial glial cells. Several transcription factors were found to be essential for the EPC-lineage differentiation, in particular, TFEB was found to tune NSC/EPC bifurcation, independent of its canonical function as a master regulator of the lysosome biogenesis. These findings suggest the potential application of TFEB-targeted clinical drugs in VZ-related disorders, such as hydrocephalus and neurodegenerative diseases.

This is a comprehensive study of the molecular mechanisms relating to VZ with potentially interesting findings. Here are some comments for improvements.

1. Compromised study design and statistical analysis are major concerns in this study. 1) No biological replicates were presented for scRNAseq and bulk RNAseq. Also, only two replicates were used in ChIP-seq and plate-based scRNAseq. Power analysis is required to estimate the sample size. 2) No statistical analysis was presented in Methods although Student's t-test was described in Fig. 5-7 legends. What about other figures (e.g., Fig. 4F, Fig. 6B-C)? It is also unclear if all the experiment groups were compared to the NC group. 3) No criteria (e.g., fold change and p value) were presented for the identification of differentially expressed genes in bulk RNAseq. 4) How were multiple testings corrected in RNAseq?

2. Why plate-based scRNAseq was used to confirm the findings of 10X scRNAseq? Is plate-based scRNAseq more reliable than 10X scRNAseq?

3. Figure 5A: The labels are inconsistent between WB and bar plot. Please check if Mcidas instead of Lhx should be labelled in WB.

4. What mouse strain was used in this study? The exact age of the neonatal mice should be provided for each experiment.

5. Rescue experiments of applying the TFEB-targeted clinical drugs to animal models with VZ-related disorders such as hydrocephalus would enhance the conclusions of this study.

---

## [Decision Letter · Decision Letter 2]

Dear Dr Zheng,

Thank you for your patience while we considered your revised manuscript "Unraveling Lineage Roadmaps and Fate Determinants to Postnatal Neural Stem Cells and Ependymal Cells in the Developing Ventricular Zone" for consideration as a Research Article at PLOS Biology. Your revised study has now been evaluated by the PLOS Biology editors, the Academic Editor and the original reviewers.

As you will see below, reviewer 2 is largely satisfied by the revision, but suggests that the sample sizes should be made more explicit in the figure legend before publication. Reviewer 1, has a number of larger lingering concerns. After discussion with the Academic Editor, we think that these remaining points can be addressed with textual changes, by tempering the conclusions in places, and making the claims more precise. Reviewer 1 highlights that some of the mechanistic data is based on ex vivo radial glial cell culture, as a limitation. While we think that additional in vivo data would strengthen the study, we would not require that for publication and think that this point can also be addressed by clearly acknowledging the limitations of the study in the discussion section. 

In light of the reviews, which you will find at the end of this email, we are pleased to offer you the opportunity to address the remaining points from the reviewers in a revision that we anticipate should not take you very long. We will then assess your revised manuscript and your response to the reviewers' comments with our Academic Editor aiming to avoid further rounds of peer-review, although we might need to consult with the reviewers, depending on the nature of the revisions.

**IMPORTANT: As you address the last reviewer points, please also address the following editorial requests: 

1) TITLE: We would like to suggest a slight tweak to your title. If you agree, please change it to "Lineage trajectories and fate determinants of postnatal neural stem cells and ependymal cells in the developing ventricular zone"

2) ABSTRACT: Per journal policy, please clearly indicate the model organism(s) studied here, in your abstract. 

3) FINANCIAL DISCLOSURES: In our editorial manager system, please update the financial disclosures statement, to include the website of each funder of your study. 

4) ETHICS STATEMENT: Please update the ethics statement in your manuscript to include the specific national or international regulations/guidelines to which your animal care and use protocol adhered. Please note that institutional or accreditation organization guidelines (such as AAALAC) do not meet this requirement.

5) DATA AVAILABILITY: : In our editorial manager system, please update your data availability statement to indicate that 'all other underlying data can be found in the manuscript and its supporting files' as this will point readers towards your source data files. Please also add a sentence to each figure legend, pointing readers to the underlying data. You can say 'The data underlying this figure can be found at ___". 

6) CODE: Per journal policy, if you have generated any custom code during the course of this investigation, please make it available without restrictions. Please ensure that the code is sufficiently well documented and reusable, and that your Data Statement in the Editorial Manager submission system accurately describes where your code can be found. 

**IMPORTANT - SUBMITTING YOUR REVISION**

*Resubmission Checklist*

*Published Peer Review*

*PLOS Data Policy*

*Blot and Gel Data Policy*

Sincerely,

Luke

Lucas Smith, Ph.D.

Senior Editor

PLOS Biology

lsmith@plos.org

REVIEWS:

Reviewer #1: Eventhough the modification in the revised version of the manuscript clarifies some of the issues raised, however there are several concerns that remain. 

The manuscript offers limited insight into the lineage choice between adult NSCs and ependymal cells as many aspects have previously reported. Several studies have performed single cell experiments in the niche (eg Neuron. 2014 May 7;82(3):545-59; Cell, Volume 173, Issue 4, 1045 - 1057.e9q; Cell Reports, Volume 25, Issue 9, 2018, Pages 2457-2469.e8). In addition the common origin of adult NSCs and ependymal cells has recently been revealed by Spassky' and Buylla's lab as well as the role of Geminin GemC1 in this fate decision.

In addition in several cases the authors conclusions lead to misconceptions (eg line 117-118 the terminology GPCs is not used by the field and it is establish that EC mature slowly losing their radial glial cell characteristics, Ascl1 is not a NSC marker as it is expressed in TAPs). 

Authors often are lead to conclusions without their experimental data to substantiate enough their conclusions (eg 166-167, 216-218, 191-194). McIdas does not suppress the conversion of GPCs into EPCs but does not allow to be fully differentiate (doi.org/10.1242/dev.172643). In addition Tfeb role is not clear as its expression is downstream of GemC1 key driver of EC commitment and differentiation and in the other hand depletion of Tfeb enhance EC differentiation (line 386). The authors do not provide solid data in order to claim "TFEB activation blocks GPCs specification in EPCs…" as they are experimental findings are based on a ex vivo Radial glial cell culture system.

Reviewer #2, Kristopher T. Kahle, MD, PhD (note, reviewer 2 has signed this review): The authors have significantly improved the manuscript by addressing most concerns from this reviewer. However, the use of only two biological replicates in single cell RNAseq remains a concern. Typically, three or more biological replicates should be used. The authors should clearly claim the sample sizes for all experiments in figure legends.

---

## [Editor Report · Decision Letter 3]

Dear Dr Zheng,

Thank you for the submission of your revised Research Article "Lineage Trajectories and Fate Determinants of Postnatal Neural Stem Cells and Ependymal Cells in the Developing Ventricular Zone" for publication in PLOS Biology and thank you for addressing the last reviewer and editorial requests in this revision. On behalf of my colleagues and the Academic Editor, Richard Daneman, I am pleased to say that we can in principle accept your manuscript for publication, provided you address any remaining formatting and reporting issues. These will be detailed in an email you should receive within 2-3 business days from our colleagues in the journal operations team; no action is required from you until then. Please note that we will not be able to formally accept your manuscript and schedule it for publication until you have completed any requested changes.

PRESS

Sincerely, 

Luke

Lucas Smith, Ph.D.

Senior Editor

PLOS Biology

lsmith@plos.org